

# Microbial Biobanking
# Cyanobacteria-rich topsoil facilitates mine rehabilitation

Wendy Williams[1], Mel Schneemilch[1], Angela Chilton[2], Stephen Williams[1], Brett Neilan[3]

1.  The University of Queensland, Gatton Campus 4343 Australia
2.  Australian Centre for Astrobiology and School of Biotechnology and Biomolecular Sciences, University of New South Wales, Sydney, NSW, 2052, Australia
3.  School of Environmental and Life Sciences, University of Newcastle, Callaghan, NSW, 2308, Australia

*Correspondence to*: wendy.williams@uq.edu.au

**Abstract**

Mining rehabilitation requires key solutions to complex issues relating to ecosystem function. In arid landscapes, the removal or disturbance of topsoil incorporating soil microbial communities can result in a shift in ecosystem function. Soil surfaces in arid regions are protected by biocrusts that regulate soil moisture, sequester carbon and fix significant quantities of atmospheric nitrogen. Cyanobacteria often dominate these bioactive surfaces and work as ecosystem engineers in that they are in sufficiently large quantities they initiate biocrust establishment and facilitate soil surface stabilisation. Cyanobacterial exopolymeric secretions form cohesive and protective layers at the soil surface that minimise wind erosion. This research encompassed soil microbial community profiling (using a polyphasic approach) with a focus on 'biobanking' topsoil for rehabilitation purposes. The research was in collaboration with Iluka Resources at Jacinth-Ambrosia (J-A) mineral sand mine located in a semi-arid chenopod shrubland in southern Australia. At J-A diverse biocrusts included a significant representation of cyanobacteria, lichens and mosses that inhabited nearly half of all soil surfaces. Cyanobacterial community structure at J-A was comprised of a variety of species having a range of attributes that contributed to their resilience and survival in an arid environment. Stockpiling from shallow scrapings and storage at low profiles appeared beneficial in microbial biobanking cyanobacterial inoculum that would facilitate recovery over time. These studies have provided information for the establishment of a monitoring program that assesses the re-establishment of biocrusts following mining. Following soil stockpiling that occurred during the mining process, cyanobacterial taxa recovered at different rates. Cyanobacterial strategies central to survival include exopolymeric production, spectral adaptation, nitrogen fixation and motility. Biocrust re-establishment during mining rehabilitation relies on the role of cyanobacteria as a means of early soil stabilisation. Provided there is adequate cyanobacterial inoculum in the topsoil stockpiles their growth and the subsequent crust formation should take place largely unassisted. Ongoing monitoring of biocrust recovery is important as it provides an effective means of measuring important soil restoration processes.



## 1.0 Introduction

Soil disturbance results in a loss of resources from arid ecosystems and often has long lasting effects on soil stability, nutrient cycling and surface hydrology (Bowker, 2007; Tongway and Hindley, 2004). Restoration of ecosystem function post disturbance requires comprehension of the dynamic functional status of the landscape prior to disturbance (Tongway and Ludwig, 1996), as well as an understanding of the net accumulative effects of disturbance on the components of that system. To evaluate the net effects of disturbance, the severity and periodicity need to be understood whereby small but frequent disturbances may have an accumulative effect, whilst rare but severe disturbance may permit natural recovery between events. Subsequently, it is necessary to appreciate the micro-processes that will assist in the restoration of soil function and to monitor their recovery along the way. Microbial communities drive micro-processes that impact on soil ecosystem function on several levels with feedbacks that can considerably assist arid landscape rehabilitation.

Cyanobacterial re-establishment is a key indicator of early soil surface re-stabilisation, regulation of soil moisture and, the balancing of soil carbon and nitrogen (Elbert et al., 2012). As biocrusts develop in structural complexity, the diversity of organisms is regulated by water infiltration from rainfall, temperature, light and disturbance (Belnap and Eldridge, 2001; Büdel et al., 2009; Elbert et al., 2012). Early successional biocrusts are mainly composed of bacteria, algae and cyanobacteria whereas late successional biocrusts can incorporate lichens, mosses, liverworts, algae, fungi and bacteria (Büdel et al., 2009). Highly structured, diverse biocrust communities are a significant asset to arid soil ecosystems providing a protective, nutrient rich layer closely integrated into the soil surface (Delgado-Baquerizo et al., 2013; Maestre et al., 2012).

In arid environments, excretions of extracellular polymeric substances (EPS) by microbes such as cyanobacteria, form stratified layers of organic and inorganic material that link and bind soil particles together (Issa et al., 2007; Rossi et al., 2017). Cyanobacteria in arid landscapes are exceptionally well-adapted to desiccation. Their polysaccharide sheaths and EPS production perform a vital role in maintaining cyanobacterial cell integrity, exchange of information and absorption of water during rehydration (Rossi et al., 2017). EPS has adhesive properties that binds non-aggregated soil particles into a protective encrusted surface that reduces the destructive impacts of wind and water (Eldridge and Leys, 2003; Rossi et al., 2017). Cyanobacterial biofilms provide initial stabilisation of disturbed surfaces that pave the way for diverse microbial communities (Büdel et al., 2009), and form bioactive crust-like layers integrated into the soil surface (Bowker et al., 2014). EPS forms the bulk of the biocrust structure. EPS more than doubles the biocrusts compressive strength and increases cohesiveness by up to six times with a ratio of at least 2:1, EPS to chlorophyll (Hu et al., 2002).

Disturbance can profoundly disrupt biocrust integrity, composition and physiological function where the impact is governed by site characteristics, severity, frequency, and timing (Belnap and Eldridge, 2001). Mining disturbance alters the biophysical state of the biocrust community through excavation, crushing, mixing and burial. The negative impacts of biocrust burial within the natural environment as a result of environmental stress and disturbance have been previously studied. For example, there were significant declines recorded in cyanobacterial diversity and abundance that was linked with a reduction in soil nutrient concentrations (Rao et al., 2012; Williams and Eldridge, 2011). Following disturbance, restoration and regrowth of





biocrusts can take place unassisted, seasonally driven and may take many years (Belnap and Eldridge, 2001; Belnap and Gillette, 1998). Alternatively, assisted biocrust restoration places emphasis on the recovery of ecosystem function and considers environmental constraints. This incorporates the knowledge of 'potential condition' based on experience with sites of ecological similarity that have undergone disturbance and recovery (Bowker, 2007). Biocrust recovery can be altered by

dust deposition, fire, and climatic conditions (Weber et al., 2016). When biocrusts recover naturally soil properties change. For example, in Southern African and Spanish rangelands an incremental accumulation of soil nutrients, organic matter and a build-up of silt and clay lead to the development of a resilient and multi-functional biocrust (Büdel et al., 2009; Maestre et al., 2012; Weber et al., 2016).

Research into biocrust disturbance and recovery post-mining is rare (see Fischer et al., 2014; Lukešová, 2001; Mazor et al.,

1996; Setyawan et al., 2016), yet there is a real need for focus on practical approaches that contribute to the restoration of soil function and measure relevant aspects of success through soil microbial communities (Harris, 2003; Tongway and Hindley, 2004). Mine rehabilitation is a complex process that involves many levels of understanding of difficult issues relating to ecosystem function where the removal or burial of the bioactive soils can have knock-on effects for rehabilitation efforts such as native seedling establishment. Successful ecological restoration of arid mining sites relies on a holistic approach where

biocrust recovery to pre-disturbance levels is integral and can serve as an indicator of the integrity of the ecosystem (Tongway, 1990).

At Jacinth-Ambrosia mine (J-A), situated on the edge of the Nullarbor Plain, South Australia, well established and diverse biocrust communities cover about 45% of the soil surfaces between plants (Doudle et al., 2011). These primarily consist of lichens, mosses and cyanobacteria. As a component of the broader landscape ecosystem biocrust communities maintain spatial

structure on the macro-scale through maintenance of landscape organisation (Bowker, 2007; Gilad et al., 2004; Tongway, 1990). On the micro-scale cyanobacterial diversity contributes to soil ecosystem function through micro-processes including carbon fixation through photosynthesis, atmospheric nitrogen fixation in a plant-available form, micro-nutrient breakdown and release, soil particle cohesion, regulation of moisture and soil surface structure (Delgado-Baquerizo et al., 2013; Elbert et al., 2012; Hu et al., 2002; Maestre et al., 2012 and others). As cyanobacterial communities often develop their diversity and

structure in response to their environment, it was important to examine this diversity on several scales.

The J-A mining process involved removal of topsoil (0–100 mm) that was either directly returned to an area under rehabilitation or temporarily stored in low stockpiles that results in biocrust burial. This crushing and burial of the biocrust leads to the inability of crust organisms to photosynthesise due to the lack of light. The effects of topsoil stockpiling on biocrust organisms such as cyanobacteria and their recovery time has not been investigated to date. Should biocrust organisms remain inactive

while they are wet, cell death and decomposition may occur (Kidron et al., 2012; Rao et al., 2012). Nevertheless, in dry conditions, cyanobacteria and algae are known to remain in a desiccated and viable for up to millions of years (Vishnivetskaya et al., 2003).

The restoration of landscape function and the accompanying need for the restoration of the soil ecosystem that included the biocrusts after high-levels of disturbance directed the development of this biocrust research project. As biocrust re-





establishment is dependent on cyanobacteria, for early soil stabilisation we needed to know the structure and diversity of the naturally occurring cyanobacterial community. It follows that cyanobacterial inoculum in the topsoil stockpiles would be central to early stabilisation of mobile surfaces subjected to the potential impacts of wind and rain splash erosion. We sought to determine whether shallow 'biobanks' of biocrust-enriched top soil would serve as viable sources of active cyanobacteria

to assist biocrust recovery of arid mining sites. The overall goals of the biocrust research program were to: (a) evaluate specific roles of natural, undisturbed biocrusts in ecosystem function at the mine site; (b) determine cyanobacterial community structure in terms of key species that drive early colonisation, biogeochemical cycling and soil stabilisation; and (c), to investigate the effects of stockpiling topsoil on cyanobacterial survival after burial and subsequent recovery. We hypothesised that cyanobacteria would be essential as ecosystem engineers in soil habitat formation with a clear distinction in community

structure between soil types and biocrust successional stages. It was also anticipated that in the topsoil stockpiles there would be a low survival rate especially at depth and over time.

## 2.0 Methods

The research program methods were conducted in two stages over a 16-month period commencing late May 2011 to Sept 2012 that was divided into several integrated studies. These studies were focused on three main levels of biocrust sampling: (a) field

sampling from 10 sites of naturally occurring biocrusts derived from the three main soil types and vegetation communities (Figs. 1, 2, Figs. S1–3, Table S1). These samples were analysed microscopically (diversity and abundance studies) and for their biophysiochemical properties; (b) field sampling of biocrusts from three contrasting successional sites (Bare, Early and Late), (Fig. 3) analysed genomically to determine early colonisers and crust building cyanobacteria; (c) sampling from six stockpiles of three different ages to determine the response of cyanobacteria to topsoil stockpiling over time. This both

underpinned the detailed analyses of their ecological attributes and provided a framework for understanding their viability and responses to disturbance including burial.

### 2.1 Field Methods

#### 2.1.1 Soil habitat and crust morphology

Sites were selected from the main vegetation associations across the three soil types and a salt lake (Fig. 1). To characterise

the cyanobacterial communities, sample sites were selected to encompass the three soil management units or SMU (Table S1), and five crust types (Table S2). A total of ten sites were selected (Fig. 1) and photographs were taken at the time of sampling (Fig. 2, Figs. S1–3). Taking into consideration there would be large areas of rehabilitated mining surfaces in the future, Site 6 (a two-year old topsoil stockpile) was regarded as an early successional biocrust that may represent a point in time (two years) since disturbance (Fig. S3).





*2.1.2 Field biocrust sampling*

Preliminary identification of biocrust types had been determined by Doudle et al., (2011) and provided the baseline data for biocrust sampling from Sites 1–10 (Fig. 1, Table S1). Within each area eight x 10 cm diameter samples were selected at random and removed to a depth of 1 cm using metal scraper (n=80), air dried (>40°C), and stored in Petri dishes. The samples were

packed to avoid crust disruption and transported to The University of Queensland's Central Analytical Laboratory at Gatton. For genomic profiling of naturally occurring successional biocrust communities, a location adjacent to Site 9 was visually determined to contain Bare, Early (Types 1-2) or Late (Types 3-5) stages of development (Table S2). Biocrust successional features were determined by morphological attributes of pigmentation, thickness and surface roughness as well as the presence/absence of lichens and mosses (Fig. 3), (Chilton et al., 2017). Bare stage was defined by loose soil particles with no

biocrust structure. Samples were collected in July 2014. For each successional stage, three replicates were collected that were representative of SMUs 1–3 where a 10 cm$^2$ plot with 95% coverage of the desired biocrust stage was excised to the depth of the crust and non-aggregated soil discarded (Fig. 4).

*2.1.3 Topsoil stockpile sampling*

Sampling was carried out in March 2012 with samples sourced from topsoil stockpiled from areas with *Acacia papyrocarpa*

(Western Myall) over-storey (Table S3). A total of six stockpiles established over three different years, detailed as months since storage in results (n=6 per time period), were sampled in triplicate to ascertain any changes in cyanobacterial survival that may occur with stockpile age. Holes were excavated by shovel using a bar to break up compacted soil. Holes were dug to a depth > 50 cm with samples taken at six depths (0–2, 2–4, 4–6, 10, 25 and 50 cm) throughout the exposed soil profile using a geologist's hammer and a spatula. Care was taken to avoid contamination by soil from the upper profiles by removing

any loose dust using a brush. A second sample at the depth of 50 cm and later autoclaved to serve as a culturing control. Corresponding samples were taken from the same depths in adjacent undisturbed areas correlated to relevant SMUs. These samples were used to identify cyanobacterial viability at depth in undisturbed areas. Samples were stored in paper bags and processed at The University of Queensland. This sampling provided n=18 for each depth of the stockpiled and undisturbed soil.

**2.2 Laboratory**

*2.2.1 Biocrust physicochemical properties*

Duplicate sub-samples derived from the ten sites were fine sieved (1.70 mm) and analysed for total C and N and C:N ratio. Analysis was carried out using a high temperature digestion in a vario MACRO Elemental Analyser (Elementar).
Soil pH and electrical conductivity (EC) were prepared with duplicate samples using a 1:5 (soil to water) ratio and shaken for

one hour. Following shaking samples were left to stand for 30 mins and then EC was measured; the sample was then mixed





again, and pH was measured. pH measurements were taken with a TPS pH meter MC-80 using an ionode IJ44C electrode. EC was measured using a Crison Conductivity Meter 525.

Chlorophyll concentration of the biocrusts were made following resurrection. The chlorophyll concentrations of the cyanobacterial crusts were determined through the extraction of chlorophyll *a*. This was carried out using a 1:5 ratio of dry biocrust to Dimethyl sulfoxide (DMSO) (Barnes et al., 1992) with samples placed in a warm bath (65°C) for a two-hour dark extraction, followed by centrifuging for five minutes (5000 g RCF). Chlorophyll *a* concentration was calculated after Wellburn, (1994).

A pocket penetrometer (8 mm foot) was used to determine the compressive strength (kg cm$^{-2}$) of the dry intact biocrusts. Four measurements were taken from each sample, providing 12 replicates per site. The measurement was taken at the point when the crust was broken, and the foot penetrated the soil surface.

### 2.2.2 Photosynthetic activity

Photosynthetic performance (recorded as yield, YII) was measured using pulse-amplitude modulated (PAM) fluorometer (Pocket PAM; Gademann Instruments, Germany). The aim was to demonstrate YII of the biocrusts using the detection of chlorophyll fluorescence from photosystem II (PSII). This is achieved by short pulses of excitation light at high intensity that is amplified resulting in a brief closure of PSII and the measurement of fluorescence yield based on the Genty parameter which is the quantum yield (YII) of the charge separation of PSII (Genty et al., 1989) and recorded on a scale of 0–1 for all photosynthesis.

### 2.2.3 Genomic profiling of bacterial community

Each biocrust replicate for Bare, Early and Late stages of development were homogenised and genomic DNA extraction performed using the FASTDNA Spin Kit for Soil (MP Bio Laboratories, USA) according to the manufacturer's instructions. Molecular libraries of the 16S rDNA V123 hypervariable region generated via PCR were submitted to the Ramaciotti Centre for Genomics (UNSW, Australia). Sequencing data was processed using Mothur version 1.34.0 (Schloss et al 2009) and described in detail in Chilton et al., (2017). Singleton and doubleton OTUs were removed and samples rarefied to 8598 sequences each across 3785 OTUS. The curated Greengenes database (McDonald et al 2012) was used to assign taxonomy to OTUs.

### 2.2.4 Cyanobacterial community structure

Three Petri dishes from each of the ten field sites were selected for incubation in the laboratory glasshouses. These samples provided fresh cyanobacterial inoculum for the range of ecosystem function studies. All samples were resurrected in the glasshouse for three to five days. Following resurrection each of the samples were analysed microscopically. Diversity was assessed with light microscopy using multiple wet mounts. In this study community diversity and structure were approached





on two scales. Each of the ten sites had field site replicates (n=30) were all divided into four microsites (n=120) which provided 12 replicates per site. A minimum of two wet-mount slides incorporated six representative portions of the cyanobacterial colonies. Therefore, cyanobacterial colonies examined for each of the ten field sites totalled 144 (12 reps. x 12 colonies, total n=1,440). For the dominant land type, Chenopod shrubland (Site 8), there were an additional 10 x 6-cell multi-well plates.

These were treated similarly where two slides were examined from each of the 60 multi-wells (n=120). In total > 2,184 cyanobacterial colonies were examined.

Initial inspection of the biocrust and the separation of individual species were made using an Olympus SZH10 microscope at 70 x magnification. Cyanobacterial filaments or colonies were carefully extracted with forceps to recover sufficient material that included important morphological features such as their colour, encasing sheaths as well as cellular structure. Live material

was examined by Nomarski differential interference contrast (DIC) microscopy with a Jenaval (Jena Zeiss) and an Olympus BX51 compound microscope (magnifications 400–1000 x). Photomicrographs were taken using an Olympus SC100 digital microscope camera, and morphological measurements of vegetative cells were made from digital images of live material taken at 400 x magnification using Olympus cellSens® digital imaging software.

Identification was performed to a species level (wherever possible) in the laboratory using the following taxonomic references:

Anagnostidis and Komarek, (2005, 2005); Sant'Anna et al., (2011); Skinner and Entwisle, (2002). It was often necessary to record the closest named species as attributes varied somewhat to temperate climate and aquatic specimens described in literature. Nitrogen fixing cyanobacteria were identified based on the three recognised types: (1) heterocystous species (those with specialised N-fixing cells); (2) non-heterocystous species that fix N aerobically and; (3) non-heterocystous species that fix N anaerobically (Bergman et al., 1997; Stal, 1995).

Using a graticule, abundance was ranked on a scale of 1-8 where the main taxa are ranked in decreasing order of the relative percentage area occupied in a single view (Biggs and Kilroy, 2000). More than one species could be dominant, and all other taxa were ranked in relation to the dominant taxa as abundant, common, occasional and rare.

### 2.2.5 Stockpile sample preparation and incubation

Samples were weighed into individual plastic Petri dishes with 20.00 g per sample. To avoid contamination, the deepest

samples (50 cm) were weighed first as the shallowest samples (0–2 cm) were considered the most likely layer to contain the highest concentration of micro-organisms. Implements were sterilised after use, the balance and surrounding area were wiped regularly with 70% methylated spirits, and hands were washed between profiles.

After weighing, approximately 9 ml of sterile distilled water was added to each sample. Samples were then sealed with parafilm. Control samples taken from 50 cm were autoclaved at 100ºC and 121 psi for 20 minutes in glass Petri dishes that

were wrapped in aluminium foil. These samples were then transferred to plastic Petri dishes and watered and sealed as per the other samples. Petri dishes were stored at 26°C under a 12-hour photoperiod regime and rotated weekly to prevent site specific effects. Samples were maintained as wet by the addition of sterile water within a laminar flow cabinet.





*2.2.6 Identification of cyanobacteria from stockpiles*

After six weeks of incubation cyanobacteria present in the samples were identified as morphotypes based on keys and methods in Section 2.2.4. Each sample was wet-mounted and examined at 16 x magnification to locate cyanobacterial thalli and colony size was estimated via area of coverage of the field of view. Where multiple morphotypes were present in a colony, the microscopic relative abundance of each was used to divide the cover between the morphotypes accordingly. Where no growth was observed, five soil samples were taken randomly from the sample, mounted and examined under 400 x magnification for the presence of cyanobacterial cells. All controls were examined first followed by undisturbed samples starting with the deepest samples then sequentially through the remaining samples from deepest to surface layers. The stockpile samples were then examined in the same order.

*2.2.7 Statistical analysis*

To determine similarities between cyanobacterial communities a cluster analysis and nMDS were conducted using Primer v6 (Clarke & Gorley 2001) and that the SIMPROF routine was used to determine significance between clusters.

For the genomic profiling of Bare, Early and Late stage communities, diversity values were derived using the DIVERSE function within the Primer package (Anderson et al 2008). ANOVA with post hoc Tukey's tests was used to test for significant differences between stages. Multivariate analyses were performed in Primer upon a Bray-Curtis dissimilarity matrix generated from square-root transformed abundance data. Samples were represented in two and three-dimensional space within a non-metric multidimensional scaling plot (nMDS). Pair-wise, a posteriori comparisons of factor Stage were performed using the PERMANOVA function with 9999 Monte Carlo permutations. Homogeneity of dispersion for each stage was tested using PERMDISP.

For the stockpiles ANOVAs were performed with post hoc Tukey's tests performed to determine any differences between treatments.

**3.0 Results**

**3.1 Soil physicochemical environment**

Soil physicochemical descriptions are provided in Table 1. Soil pH across all sites ranged from 8.4–8.6 and the topsoil stockpile was higher at 8.9. Electrical conductivity ranged from 92–140 µS cm$^{-1}$. There were no ecologically significant differences in pH and EC between field samples and intact samples (data not shown). Total nitrogen was typically <0.1% across all sites and total carbon ranged between 1–2% with higher percentages generally found across SMU 2 and 3. The ratio between carbon and nitrogen was the greatest across SMU 3 and the topsoil stockpile, also originating from SMU 3. This was as a result of a higher percentage of soil carbon in those sites.




For the intact biocrusts mean chlorophyll concentrations ($\mu$g g$^{-1}$ soil) differed across all sites and SMU 1 biocrusts were significantly lower (p = 0.05) compared to SMU 2 and 3 (Fig. 4). The mean chlorophyll concentrations of biocrusts sourced from the 2-year old topsoil stockpile (7.49 ± 1.01 $\mu$g g$^{-1}$ soil) were almost half the concentration of SMU 3 (13.53 ± 1.74 $\mu$g g$^{-1}$ soil), which was the origin of the topsoil.

The penetrometer index of crust compressive strengths across the ten sites showed that Sites 4, 9 and 10 in SMU 1 were significantly different from each other as well as significantly lower than SMU 2, SMU 3 and T2 sites (Table 2). The means for SMU 1–3 were also significantly different (p < 0.0001), (Levene's test), and it was confirmed by a student's t-test that the difference was between SMU 1 and SMU 2–3.

### 3.2 Photosynthetic yield

The photosynthetic yield measurements showed there was variability between sites (Fig. 5). The yields recorded all fell within the expected range and when data was separated into their respective SMU it provided a clearer picture of the variability. SMU 3 had the lowest mean overall while there were no differences between the others.

### 3.3 Successional diversity

Microbial community profiling using high through-put sequencing revealed cyanobacteria comprised a significant component
of all three stages, forming the majority of sequences in Early and Late stages (Fig. 6). There was a diversity of morphotypes observed including simple filamentous, heterocystous and unicellular types (Fig. 7). The most abundant genera identified were *Leptolyngbya, Phormidium, Tolypothrix, Nostoc, Brasilonema, Chroococcidiopsis* and *Acaryochloris*. Unclassified Nostocaceae were dominant within Bare soils while the Early stage was more even. This was due to an increase in *Phormidium, Brasilonema* and the unicellular genera. Late stage biocrusts observed a slight resurgence of Nostocaceae but was still
relatively even. There was no significant difference in the richness, evenness or diversity between the three stages (Table 3). However, there were significant differences in the composition and structure of the communities of each stage. The Bare stage was significantly different to the crusted stages (Fig. 8, Table 4) while resolution between the Early and Late stages was less clear. Ordination of the samples within three dimensions showed Early and Late stages grouped separately however, PERMANOVA failed to identify a significant difference (Table 3). Greater sampling may provide stronger discerning power
between Early and Late stages in future work to determine whether these communities are microbiologically distinct.

### 3.4 Cyanobacterial community macrostructure

Across the 10 sites examined, the biocrusts examined were made up of both surface and subsurface cyanobacteria. The biocrust was stratified into surface and subsurface layers that comprised of a closely interwoven network of sticky mucilaginous filaments that bound together the soil particles (Fig. 9a–f). The biocrust depth ranged from 1–10 mm and could be easily
removed from the soil surface and held as a separate biophysical structure (Fig. 9f).



### 3.5 Cyanobacterial community microstructure

A total of 21 cyanobacterial species were identified via microscopy across the ten sites (Table 5; Figs. S8–15). The majority of the diversity and abundance comprised of four filamentous genera: *Symploca* (18%), *Schizothrix* (16%), *Porphyrosiphon* (16%) and *Scytonema* (16%). Secondary cyanobacteria present variously occupied <1 to 10%, and overall made up 34% of

the community.  The known nitrogen-fixing cyanobacteria *Symploca, Scytonema*, *Porphyrosiphon*, *Brasilonema, Nostoc* and *Gloeocapsa* comprised more than 50% of the diversity at each site and formed 61% of the total community diversity (Fig. 10). In this study cyanobacterial community structure was tested against the three different soil types to determine whether soil type was influential in determining community structure. There was no significant difference in cyanobacterial community structure across soil types however; the results do suggest some spatial structuring exists across the three SMU (Figs. 11, 12).

The structural relationship between all samples shows greater similarities rather than dissimilarities, of which there are only a small number of samples that are significantly different (Fig. 12). Soil type did not explain these differences. This suggests that most of the species could potentially be found anywhere across the three zones and their diversity and abundance is controlled by other factors. *Symploca* occurred more frequently and was more abundant in the majority of the samples examined therefore was the most significant contributor to the community (data not shown). Individual sites generally

displayed similar trends although there was some variability occurring between sites.

Cyanobacterial crusts from the dune regions on SMU 1 (deep calcareous yellow sands) were representative of crust types 1–3; patchy, brittle (when dry) early-successional crusts as well as formed dark crusts that were mid to late-successional and included cyanolichens (also see Doudle et al., 2011). There were 12 cyanobacteria identified in SMU 1 where four primary genera made up 75% of the community. These cyanobacteria were: *Symploca* (26%); *Schizothrix* (20%); *Scytonema* (16%)

and *Symplocastrum* (13%). Cyanobacteria with the capacity to fix nitrogen (*Symploca*, *Scytonema*, *Porphyrosiphon* and *Brasilonema*) contributed to 55% of the community structure (Fig. S4).

Cyanobacterial crusts from the chenopod shrublands and open woodlands in SMU 2 (shallow calcareous sandy loam) represented a broad range of crust types (2–5) but overall could be described as late-successional. Lichens and mosses were highly visible (also see Doudle et al., 2011). There were 21 cyanobacteria recorded: four were primary genera that made up

63% of the community including: *Schizothrix* (17%); *Porphyrosiphon* (16%); *Scytonema* (16%) and *Symploca* (14%). Cyanobacteria with the capacity to fix nitrogen (*Symploca*, *Scytonema*, *Porphyrosiphon*, *Brasilonema, Nostoc* and *Gloeocapsa*) contributed to 60% of the community (Fig. S5).

Cyanobacterial crusts from the open woodlands in SMU 3 (deep calcareous sandy loam, Fig. 2c) represented a broad range of crust types (2–5) but like SMU 2 could also be described as late-successional. Lichens and mosses were highly visible (see

Doudle et al., 2011). There were nine cyanobacteria recorded of which four were primary genera that made up 85% of the community. These cyanobacteria were:  *Symploca* (32%); *Porphyrosiphon* (24%); *Scytonema* (19%) and *Schizothrix* (10%). Cyanobacteria with the capacity to fix nitrogen (*Symploca*, *Scytonema, Porphyrosiphon* and *Brasilonema*) contributed to 77% of the community structure (Fig S6).





Cyanobacterial crusts from Site 6 were from the 2YO topsoil stockpile that had originated from SMU 3 (deep calcareous sandy loam) would be described as early successional crusts with some seasonal mosses. There were eight cyanobacteria recorded of which four were primary genera that made up 84% of the community (Fig. S7). These cyanobacteria were: *Symploca* (25%); *Symplocastrum* (25%); *Porphyrosiphon* (24%); *Scytonema* (10%). It was interesting to note that *Symplocastrum* was co-

dominant with *Symploca* whereas in the other communities it ranged between 8-13%. Sub-surface species *Schizothrix* only contributed to 4% of the diversity compared to 10-20% elsewhere. Cyanobacteria with the capacity to fix nitrogen (*Symploca*, *Porphyrosiphon*, *Scytonema* and *Brasilonema*) contributed to 61% of the community structure.

## 3.6 Topsoil stockpiles

### 3.6.1 Morphotype differentiation

A total of nine cyanobacterial morphotypes were identified from the samples: all but one were filamentous. *Nostoc, Scytonema, Microcoleus, Porphyrosiphon* and *Leptolyngbya* were the most widespread. *Leptolyngbya* was present as both a common green and yellow morphotype and a less commonly encountered for black morphotype. *Porphyrosiphon* was also present in yellow and green morphotoypes and two distinct *Nostoc* morphotypes were observed. The least frequently sampled morphotype was *Stigonema*.

### 3.6.2 Cyanobacterial diversity

Average morphotype diversity was highest in stockpiled samples at and above 10 cm depth for all stockpile ages (Figs. 13, 14). These differences were significant at 50 cm in all stockpiles (29-month DF 5, F 7.00, P 0.024 20-month DF 5, F 8.37, P 0.016 9-month DF 5, F 18.00, P 0.002) and 25 cm in 20 and nine-month old stockpiles (DF 5, F 32.73, P 0.000 DF 5, F 16.20 respectively). Conversely, after nine and 20 months of stockpiling, average morphotype diversity was higher in stockpiled

samples when compared with undisturbed samples except for 20-month old 4–6 cm samples (Figs. 13 a, b) but this difference was only significant in the nine-month old stockpile at 10 cm depths (DF 5, F 8.27, P 0.017). High variability between replicates accounted for the lack of significance.

When comparing average morphotype diversity at different depths between stockpiles, diversity was greater in material stockpiled for the least amount of time above 10 cm depth but not at or below this point (Fig. 13a). Average morphotype

diversity was variable within adjacent undisturbed areas (Fig. 13b). The variability in morphotype diversity within replicates was high as shown in the error bars of Figures 14a–c. This variability was evident in the lack of significant difference in species diversity at any depth between stockpiles of different ages. The area covered by each of the morphotypes was relatively constant between sites in the undisturbed samples, but in stockpiled samples *Nostoc* cf *commune* was more prevalent (DF 5, F 5.97, P 0.012).

*Nostoc* cf. *commune, Nostoc* yellow, *Microcoleus* and *Leptolyngbya* were present in more stockpiled samples than undisturbed samples (Fig. 15). Conversely, *Scytonema* and the black form of *Leptolyngbya* were more prevalent in undisturbed samples

(Fig. 16). When all samples were combined, *Nostoc* cf *commune* covered the greatest area in both stockpiled and undisturbed samples followed by yellow *Nostoc* in stockpiles only. The area covered by the remaining morphotypes was similar for most other morphotypes although *Microcoleus* and *Leptolyngbya* covered a greater area in stockpiled areas and *Porphyrosiphon* covered slightly higher area in undisturbed samples. The cover of *Scytonema* was almost identical in stockpiled and undisturbed areas and this lack of difference was supported statistically (DF17, F 0.00 P 0.969).

Contamination occurred in seven of the autoclaved controls. Only one morphologically distinct *Nostoc* morphotype grew in the control samples. The contamination source could be traced back to a single watering event. Fungal growth was noted only on control samples.

### 3.6.3 Cyanobacterial distribution

*Nostoc* cf. *commune* exhibited the best survival response in stockpiling as it covered a significant area of samples from all depths in all stockpile ages. This morphotype covered the largest area of the samples in the oldest stockpiles followed by the most recently created stockpiles. These differences were significant from depths of 6 cm to the surface (4-6 cm DF 5, F 28.83, P 0.000 2-4 cm DF 5, F 4.89, P 0.023 0-2 cm DF 5, F 3.72, P 0.049). These patterns are not reflected in the total area covered by *N.* cf. *commune* in corresponding undisturbed areas. The *Leptolyngbya* black morphotype only occurred between 20 and 40 cm depth and were found in only one stockpile samples but were present in six samples from three adjacent areas. The *Stigonema* morphotype was found in six stockpile samples spanning all ages and in only one adjacent sample in all cases in the upper 10 cm of the soil profile.

An unknown red non-filamentous microbe was present in only one of the undisturbed samples but observed in stockpiles at various depths. This morphotype was found in all stockpiles except those created five months prior to sampling.

### 3.6.4 Growth rates

Growth rates are described in Figure 17 with *Nostoc* cf. *commune, Porphyrosiphon, Microcoleus* and *Scytonema* the first morphotypes to develop to an identifiable stage. Filaments of the *Stigonema* morphotype were found in low numbers and appeared to be recently formed. It was only present in samples examined in the latter stages of the identification process. The yellow form of *Nostoc* exhibited a much slower rate of development than *Nostoc* cf *commune* and could only be definitively determined as a form of *Nostoc* when examined after 13 weeks of incubation.

Samples from undisturbed areas sourced at depths from between 10 cm and 50 cm initially showed no visible signs of growth when examined after six weeks of incubation. When re-examined six weeks later cyanobacterial growth was evident but in many cases, had not advanced to the point that most of the morphotypes could be distinguished. However, the few colonies that had grown to the point that morphotypes could be distinguished included each of the morphotypes that were present in the samples from the surface layers except for the red non-filamentous organism, the black form of *Leptolyngbya* and the *Stigonema* morphotype. The results of the initial examinations were included in analyses to avoid skewing the data by including growth from a much later date.





## 4.0 Discussion

### 4.1 Microbial biobanking

Cyanobacterial community structure at J-A was comprised of a variety of species having a range of attributes that contributed to their resilience and survival in an arid environment. Stockpiling from shallow scrapings and storage at low profiles appeared beneficial in microbial biobanking cyanobacterial inoculum that will recover diversity over time. These studies have provided information for the establishment of a monitoring program that assesses the re-establishment of biocrusts following mining. Type 1–2 biocrusts would be indicative of a stable early successional cyanobacterial biofilm consisting of crust building microbes that would provide important contributions to the soil ecosystem including microorganism diversity, stabilisation, carbon and nitrogen cycling and soil surface protection (Fig. 18). The important parameters include: (a) photosynthetic yield via non-destructive field tests that verify biocrust development by microbes that fix $CO_2$; (b) biocrust compressive strength via non-destructive field tests (using a penetrometer) that establish the relative strength of the re-established biocrust; (c) cyanobacterial cover: non-destructive field tests that establish the regrowth of surface dwelling species; and, (d) cyanobacterial chlorophyll; destructive tests that determines the chlorophyll concentration of the cyanobacterial component of the biocrust including sub-surface species. At J-A once Type 1–2 biocrusts have re-established temporal biocrust monitoring could be incorporated into the Landscape Function Analysis program (Tongway and Hindley, 2004) that includes a measure of biocrusts in its raw indicator set. Type 1–2 biocrusts would be indicative of a stable early successional cyanobacterial crust that would provide important contributions to the soil ecosystem including microorganism diversity, stabilisation, carbon and nitrogen cycling and soil surface protection. Yet, for early successional biocrusts to be integrated into the general methods of LFA several studies across a multiple soil types would need to be carried out.

In summary:

1. On the broad scale cyanobacterial diversity and abundance was not affected by any differences in the local soil types.
2. Primary cyanobacteria that greatly contributed to crust formation and the production of EPS were: *Symploca, Nostoc, Schizothrix, Leptolyngbya*.
3. Nitrogen fixing cyanobacteria generally contributed to more than half of the diversity.
4. Cyanobacterial growth was seasonal (unpublished data) and was impacted by moisture availability, environmental stress and disturbance.
5. Biocrust re-establishment monitored over time will provide indicative measurements and information about soil function. This data may be representative of the key milestones relating to stability, nutrient cycling and infiltration (see Tongway and Hindley, 2004).





6. Cyanobacteria with crust building and nutrient augmentation attributes which could lend themselves to use as inoculum for facilitative restoration were: *Nostoc, Porphyrosiphon, Scytonema* and *Symploca*. These cyanobacteria could be tested for their potential as inoculants in future research.

## 4.2 Cyanobacterial endurance

Environmentally induced strategies of arid land cyanobacteria reflect their habitat, these survival traits have developed over a long evolutionary history. Many primary (common to abundant) and secondary (uncommon) cyanobacteria recorded at J-A exhibited thick gelatinous sheaths (*Porphyrosiphon, Schizothrix, Microcoleus, Nostoc*) or were associated with the production of EPS (*Symploca, Nostoc, Schizothrix, Leptolyngbya*). Filamentous cyanobacteria formed the major part of the J-A crust structure with tufts, webs or creeping masses closely intertwined (e.g. *Porphyrosiphon*, *Symploca*, *Scytonema*, *Schizothrix*,

*Microcoleus*). These are often assimilated with unicellular forms (e.g. *Gloeocapsa*, *Chroococcus*, *Chroococcidiopsis*) or gelatinous colonies of *Nostoc.*

Cyanobacteria have the capacity to tolerate high UV and light intensities, low water potential and desiccation, they are adapted to wet-dry cycles and drought, high salinity and the optimisation of short growing seasons (Büdel et al., 2009; Garcia-Pichel and Castenholz, 1993). The variability in cyanobacterial diversity recorded across the J-A landscape seemed to be mostly a

function of the arid environmental conditions. The relative abundance of different cyanobacterial genera fluctuated seasonally and was altered by salinity gradients, environmental stress and disturbance (unpublished data).

Biocrust functional groups can be classified into habitats with similar soil chemistry and texture (Bowker et al., 2011). The main soil types at J-A were calcareous in nature with some variability in texture particularly in the dunes however, the soil pH was governed by calcium carbonates and only varied across a relatively narrow range (8.4–8.9). Soil particle size can influence

crust type, community structure and biocrust establishment (Bowker, 2007; Büdel et al., 2009; Hu et al., 2002; Rozenstein et al., 2014). This is consistent with the relevance of crust types that were used as a sampling strategy in these studies.

The cyanobacterial diversity at J-A was determined according to their morphological features. In many cases these features (e.g. outer protective sheaths, UV protection, EPS production) provided the basis of attributes that pertained to fundamental survival strategies. Twenty-one cyanobacteria were recorded from 13 genera. Four species were unicellular and the remaining

seventeen were filamentous. Some cyanobacteria found at J-A (*Microcoleus paludosus, Nostoc* sp*., Gloeocapsa*) had also been recorded at Lake Gilles (SA) about 400 km southeast of J-A (Ullmann and Büdel, 2001). The taxonomic status of *Brasilonema* remained uncertain and may be a variety of *Scytonema*, however, genomic data supported morphological identification and the type has also been recorded in other terrestrial habitats globally, and due to its similar morphological attributes was called *Brasilonema* in this study (Fiore et al., 2007; Vaccarino and Johansen, 2012).

*Nostoc commune* var. *flagelliforme* had been recorded at J-A along with *Nostoc commune* across the shallow and deep sandy loams. Although *N. flagelliforme* appeared rarely, it had been previously documented from sites in south-western South Australia, Western Australia and the Northern Territory (Skinner and Entwisle, 2002) and Victoria (W. Williams, unpublished



data). Due to its scarcity *N. flagelliforme* is of high commercial value in Asia where it is considered a delicacy (Gao, 1998). A joint Spanish-Australian study has now shown that both *Nostoc commune* and *N. flagelliforme* contain the same genomic markers and cannot be separated, rather the spaghetti-like tubes that are unique ecotype likely associated with aridity (Aboal et al., 2016). It may be more widespread in Australia than previously recorded as it is often only clearly visible following rains.

## 4.3 Biocrust development

In a desert environment where moisture availability is intermittent and unpredictable, it is a highly limiting factor, and the period of time for biocrust structure to develop to a level of stability could be in reality much longer (Belnap and Eldridge, 2001). Still, cyanobacteria are well adapted to optimising seasonal conditions where there are only small windows of opportunity for growth (Büdel et al., 2009). Terrestrial cyanobacteria use oxygenic photosynthesis to drive $CO_2$ fixation substantiated though the positive photosynthetic yield recorded in all J-A biocrusts. Carbon uptake by cyanobacteria primarily results in the formation of structural biomass and organic matter (Stal, 2003). Chlorophyll concentrations can be indicative of the presence of an organic layer (EPS) that incorporates organic and inorganic materials (Davey and Clarke, 1992; Hu et al., 2002). Biocrusts need to reach a defined concentration of chlorophyll concentration before cyanobacterial secretion of EPS occurs, and then it can take several weeks before the inorganic layer is added (Hu et al., 2002). The contribution of EPS to crust structure can be measured by crust compressive strength, which also relates well to crust type, biocrust development and the quality of the binding of soil particles (McKenna Neuman and Rice, 2002; Thomas and Dougill, 2007; Wang et al., 2006). At J-A seasonal carbon and nitrogen were always lower in the biocrusts from the dunes compared to those from the other soil types. Should carbon to nitrogen ratios exceed 30, then nitrogen is likely to be a limiting factor in microbial activity (Kaye and Hart, 1997). Due to the large proportion of nitrogen fixing cyanobacteria present at J-A, if they are not restored into the system nitrogen depletion and the imbalance of the C:N ratio may also impact plant reestablishment (Evans and Ehleringer, 1994; Hawkes, 2003). In the J-A studies it appeared that the functional role of cyanobacteria in carbon and nitrogen fixation maintained soil carbon to nitrogen ratios at an optimum range for microbial activity. Carbon and nitrogen values paralleled the J-A biocrust types and, could be likened to examples of African desert crusts of similar developmental stages (Büdel et al., 2009).

Cyanobacteria that have the capacity to biologically fix atmospheric nitrogen were highly represented in the J-A biocrust diversity. Overall N-fixing cyanobacteria contributed to at least 61% of the community abundance (*Symploca, Scytonema, Porphyrosiphon, Microcoleus, Brasilonema, Nostoc* and *Gloeocapsa*) while across the different soil types including the stockpile crusts it ranged between 55–77%. It has been well documented that nitrogen fixed by both cyanobacteria and cyanolichens is freely available to higher plants and therefore an important source of plant-available nitrogen (e.g. Aranibar et al., 2004; Evans and Ehleringer, 1994). When it rains, plant-available nitrogen is flushed out of cyanobacterial sheaths into the surrounding soil, where it can be taken up by plants (Dojani et al., 2007). In arid landscapes micro-scale surface variability favours seed establishment (Eldridge et al., 1991), and there exists a plant dependence on cyanobacterial mediated nitrogen





sources (Dojani et al., 2007; Rogers et al., 1966). To this extent, the high representation of nitrogen-fixing cyanobacteria found at J-A, further supports previous findings from similar landscapes.

## 4.4 Cyanobacterial communities

This research demonstrated that the cyanobacterial communities in the J-A biocrusts were compositionally distinct topsoil
microbiomes (Fig 7). Key cyanobacteria indicating biocrust formation and development were *Leptolyngbya, Phormidium, Tolypothrix, Nostoc, Brasilonema, Chroococcidiopsis* and *Acaryochloris* (Fig. 8). These genera have consistent morphological traits with those observed via microscopy. Notably, the identification of *Brasilonema* was supported with sequencing data. Simple filamentous types are often attributed with the primary crust building role, able to span inter-particle gaps within the soil via supra-cellular structures (Garcia-Pichel and Wojciechowski 2009). Sequencing data showed *Phormidium* was the
dominant cyanobacterium for this role and it is likely that *Symploca* identified though microscopy was the principal *Phormidium* present. *Microcoleus* sp. and *Porphyrosiphon* were also identified as early colonisers however these genera are currently poorly resolved phylogenetically (Garcia-Pichel et al 2013) but share critical morphological features enabling biocrust formation and maintenance. Diversity indices derived from sequencing data of the whole bacterial community are poor measures of biocrust formation and development. Detecting increases in key species and shifts of community structure
will likely provide more informative and robust verification of desired rehabilitation outcomes.  Increased cyanobacterial biomass is likely to also be a good indicator and reliable metric.

We had hypothesised that cyanobacteria would be central to soil micro-processes which was strongly supported by the diversity and resilience of the species identified. Contrary to our hypothesis there was not always a clear distinction in community structure between soil types (Figs. 11, 12) and biocrust successional stages, notably in the early and late stages (Fig 8). Yet,
even though study cyanobacterial community diversity and abundance were not affected by soil type it appeared SMU 1 (early stage) and SMU 3 (late stage) were similar (Fig. 11) which was further supported though the sequenced samples (Fig 9) where there were no significant differences. At J-A any of the cyanobacteria could conceivably occur anywhere on the macro-scale, with their relative abundance likely determined by microenvironments and microhabitats. On a small scale cyanobacterial diversity and abundance is dictated by factors such as light (sun and shade) and chemical gradients (Stal, 2000), as well as
moisture availability and soil particle size (Büdel et al., 2009). On the broader landscape scale small-scale variations are integrated into overall community diversity and abundance. Additionally, when the biocrusts were assessed as a whole community (including lichens and other microbes), Doudle et al., (2011) showed that J-A crust types were defined by their developmental or successional stage, differed across soil types. Other studies also show that soil type can influence successional stage (Bowker, 2007; Büdel et al., 2009; Dojani et al., 2011) however as cyanobacteria were focused on in these
studies, differences in later successional crusts types that incorporate a substantial lichen and moss component were not discernible.





### 4.5 Cyanobacterial responses to stress and disturbance

Micro-scale physicochemical gradients within cyanobacterial communities' force different functional groups to inhabit niches that they have adapted to (Stal, 1995, 2003). Desert cyanobacteria are subject to high weather-induced stress but typically, low in physical disturbance. Cyanobacteria have adapted to extremes of temperature, irradiation and low moisture availability,

where they endure long periods of desiccation during which time they are inactive. Nevertheless, episodic erratic environmental events can be detrimental to cyanobacterial populations, even though they have a great capacity to adapt to changes in conditions. Physical disturbance of biocrusts occurs on a large scale at the J-A mine site with the removal and temporary stockpile storage of topsoil. This type of mechanical disturbance results in the burial and translocation of the biocrust. Only a few studies have recorded the impacts of burial within the natural environment because of drought as well as

under artificially reconstructed burial trials. A study based in China showed that there were significant reductions in chlorophyll concentration, UV synthesis, total carbohydrates (EPS) and damage to photosynthetic activity (Rao et al., 2012). In a semi-arid grassland in Australia, burial of cyanobacterial crusts during a severe drought resulted in a significant reduction in surface dwelling cyanobacteria and significant reductions in plant-available nitrogen (Williams and Eldridge, 2011).

### 4.6 Cyanobacterial resilience to stockpiling

The findings of this study show that cyanobacteria can survive stockpiling for over two years. This is not surprising due to the recognised ability of cyanobacteria to survive in extreme environments. In previous studies, cyanobacteria have been grown from samples sourced at 18 cm depths in Japanese rice paddy soils (Fujita and Nakahara, 2006), 50 cm in the UK (Esmarch, 1914), and 70 cm depths in the USA (Moore and Karrer, 1919).

The species sampled at J-A have a well proven track record of survival under extreme conditions. *Microcoleus* and

*Leptolyngbya* have survived and remained viable after up to three million years frozen in lake sediments in permafrost (Vishnivetskaya et al., 2003). Vegetative *Nostoc commune* material retains viability following several decades of storage in desiccated form (Bristol, 1919; Lipman, 1941). Reactivation of vegetative material after decades of storage was successful but several months (Lipman, 1941) to a year (Bristol, 1919) of incubation can be necessary for growth to take place. These results were reflected in the current study where growth was not observed in the undisturbed areas below 10cm depth for

several months. It may be that the longer the period of inactivity, the longer time taken for reactivation to occur.

Akinetes are desiccation resistant cells produced by certain filamentous cyanobacteria that can survive for long periods. *Nostoc* and *Scytonema* can produce akinetes (Kaplan-Levy et al., 2010; Tomaselli and Giovannetti, 1993) but many of the other species sampled in this study cannot, therefore alternative survival methods are in action. Heterotrophic growth is also possible for some cyanobacteria (Flores and Herrero, 2010). Cyanobacteria can survive in darkness through utilisation of alternate carbon

sources in drinking water systems (Codony et al., 2003) and this may also be true for soil cyanobacteria (Reisser, 2007). *Nostoc* have the potential to grow at low light in caves and under ice (Dodds et al., 1995) and even in darkness (Huang et al., 1988). Belnap and Gardner, (1993) reported *Microcoleus vaginatus* sheaths at depths to 10 cm and considered the sheaths to





be remnant from a time when the surface was lower than the current day due to a lack of chlorophyll. It is possible that heterotrophic growth was still occurring at these depths for which chlorophyll is unnecessary.

The diversity in taxa at depths in undisturbed areas was like that of surface samples yet with much slower growth. The fact that these organisms took much longer to growth than those sampled from upper layers would suggest that they have grown

from vegetative material that has been photosynthetically inactive for long periods. Long term inactivity of vegetative material can result in long lag times for growth following re-activation (Bristol, 1919; Lipman, 1941; Shaw et al., 2003) and this was observed in species sourced from depths that are incapable of akinete production.

**5.0 Conclusions**

Biocrusts and cyanobacteria are a major component of the J-A landscape that protect and enhance soil function. These studies focused on the cyanobacterial community structure at J-A and its recovery following stress and disturbance. Cyanobacterial diversity is an important measure of biocrusts that incorporate micro-processes central to a healthy and functional soil ecosystem. Cyanobacteria are well adapted to long periods without water, the optimisation of short growing seasons, wet-dry cycles, low water potentials, tolerance of high UV and low light intensities, fluctuating temperatures and in some cases high

salinity. Cyanobacterial strategies central to survival include EPS production, spectral adaptation, nitrogen fixation and motility. Biocrust re-establishment during mining rehabilitation relies on the role of cyanobacteria as a means of early soil stabilisation. Provided there is adequate cyanobacterial inoculum in the topsoil stockpiles their growth and the subsequent crust formation should take place largely unassisted. Ongoing monitoring of biocrust recovery is important as it provides an effective means of measuring important soil restoration processes.





Table 1: Intact biocrust soil physicochemical descriptions for all sites EC = electrical conductivity in µS cm$^{-1}$; total percentage of nitrogen present (N%), total percentage of carbon present (C%) and carbon to nitrogen ratios (C:N) for all sites.

| Vegetation | Soil | SMU | Site | pH | EC | N% | C% | C:N |
|---|---|---|---|---|---|---|---|---|
| Mallee | Deep calcareous yellow sand | 1 | 4 | 8.6 | 92 | 0.07 | 1.02 | 15.1 |
| Mallee | | | 9 | 8.5-8.6 | 107 | 0.07 | 0.93 | 13.7 |
| Mallee | | | 10 | 8.5-8.6 | 118 | 0.09 | 1.37 | 15.8 |
| Myall | Shallow calcareous sandy loam | 2 | 1 | 8.4-8.5 | 122 | 0.1 | 1.2 | 11.7 |
| Mallee | | | 5 | 8.4 | 133 | 0.13 | 1.84 | 13.8 |
| Chenopod | | | 8 | 8.6 | 124 | 0.1 | 1.32 | 12.8 |
| Myall | Deep calcareous sandy loam | 3 | 2 | 8.5 | 113 | 0.09 | 1.09 | 12.1 |
| Myall | | | 3 | 8.6 | 114 | 0.09 | 1.99 | 21.4 |
| Myall | | | 7 | 8.5-8.6 | 140 | 0.1 | 1.76 | 18.3 |
| Myall | Topsoil stockpile | Origin SMU 3 | 6 | 8.9 | 119 | 0.08 | 1.57 | 19.6 |

Table 2: Biocrust compressive strengths measured by penetrometer: means and standard deviations (SD) for SMU 1–3 (kg cm$^{-2}$) tests with p-values, values in bold that are different from 0 with a significance level alpha=0.05, NS = not significantly different.

| SMU | Means ± SD | SMU 1 | SMU 2 | SMU 3 |
|---|---|---|---|---|
| SMU 1 | 2.79 ± 1.41 | **0** | **0.001** | **0.005** |
| SMU 2 | 3.75 ± 0.79 | **0.001** | **0** | NS |
| SMU 3 | 3.97 ± 0.70 | **0.005** | NS | **0** |

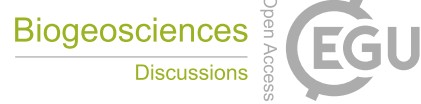



Table 3: Cyanobacterial mean (± Standard Error) of richness (Margalef's index), evenness (Pileau's index) and diversity (Shannon index) across successional stages.

| | Bare | | Early | | Late | |
|---|---|---|---|---|---|---|
| | Mean | SE | Mean | SE | Mean | SE |
| **Richness (d)** | 167.4 | 5.344 | 146.8 | 5.818 | 142.8 | 15.69 |
| **Evenness (J')** | 0.816 | 0.011 | 0.791 | 0.005 | 0.788 | 0.027 |
| **Diversity (H')** | 5.977 | 0.103 | 5.692 | 0.064 | 5.642 | 0.281 |

Table 4: Permutational analysis of variance (PERMANOVA) of pair-wise comparisons between biocrust stages and bare soil. P(MC) = probaility values obtained using 9999 Monte Carlo permutations. Permdisp showed no significant differences in variation of spread (pseudo $F$=3.8068, $P$(perm) = 0.068).

| Groups | t | P(perm) | Unique permutations | P(MC) |
|---|---|---|---|---|
| **Bare, Early** | 2.6216 | 0.0979 | 10 | 0.0107 |
| **Bare, Late** | 2.5742 | 0.0959 | 10 | 0.0120 |
| **Early, Late** | 1.2793 | 0.0953 | 10 | 0.1993 |





Table 5: Diversity across sites on a presence absence basis for all seasons and Lake Ifould (salt lake). Different species attributed to a genus (i.e. sp. 1,2,3) have all been separated based on their morphological features and size but could not be positively identified.

| | SMU 1 | SMU 2 | SMU 3 | Stockpile | Lake Ifould |
|---|---|---|---|---|---|
| *Aphanothece* | | | | | x |
| *Brasilonema* | x | x | x | x | x |
| *Chroococcidiopsis* | | x | | x | |
| *Chroococcus* sp. 1 | x | x | x | x | |
| *Chroococcus* sp. 2 | | | | | x |
| *Gloeocapsa* | x | x | | | x |
| *Leptolyngbya* | x | x | | | x |
| *Microcoleus chthonoplastes* | | | | | x |
| *Microcoleus paludosus* | x | x | x | x | x |
| *Microcoleus sociatus* | | | | | x |
| *Microcoleus vaginatus* | x | x | | | x |
| *Nostoc commune* | | x | x | x | x |
| *Nostoc flagelliforme* | | x | | | |
| *Nostoc* cf. *pruniforme* | | x | | | |
| *Nostoc* sp. | x | x | | | |
| *Porphyrosiphon* sp. 1 | x | x | x | x | |
| *Porphyrosiphon* sp. 2 | | x | | | x |
| *Schizothrix* sp. 1 | x | x | | | x |
| *Schizothrix* sp. 2 | | x | x | x | |
| *Schizothrix* sp. 3 | | | | | x |
| *Scytonema* sp. 1 | x | x | x | x | |
| *Scytonema* sp. 2 | | x | | | x |
| *Scytonema* sp. 3 | | x | | | |
| *Scytonema* sp. 4 | | | | | x |
| *Symploca* sp. 1 | x | x | x | x | x |
| *Symploca* sp. 2 | | x | | | |
| *Symplocastrum* sp. 1 | x | x | x | x | x |
| *Symplocastrum* sp. 2 | | | | | x |
| **Diversity** | **12** | **21** | **9** | **10** | **18** |



Figure 1: Image of biocrust sample sites located within the vegetation associations described in Table S1 (supplied, S. Doudle).





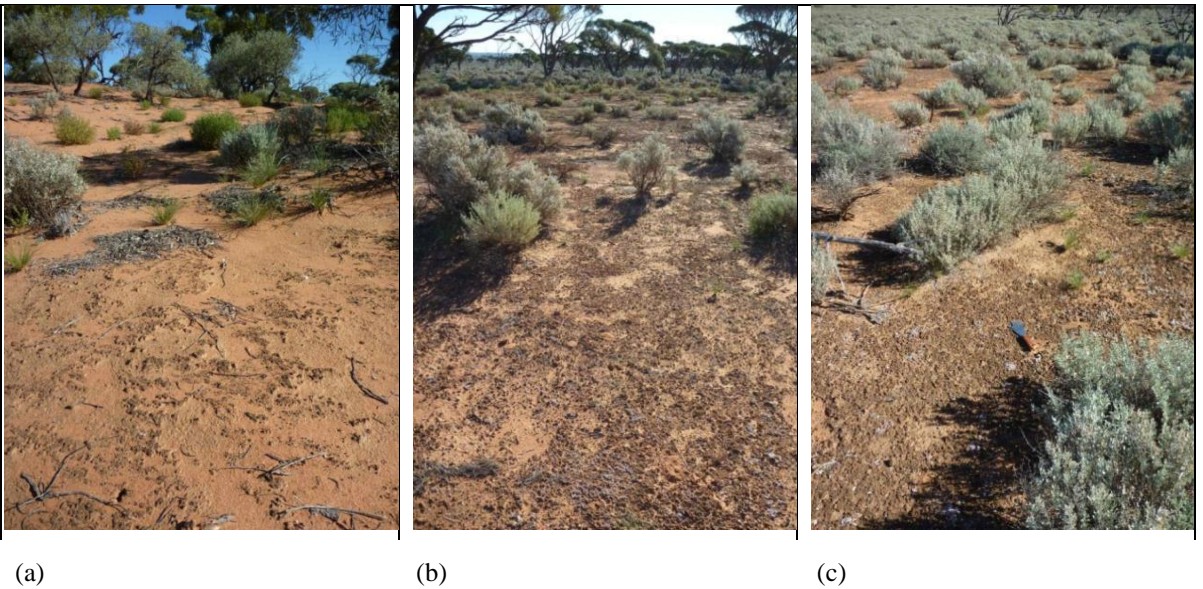

   (a)                                    (b)                                    (c)

Figure 2: (a) SMU 1: Type 1-3 Biocrusts on deep calcareous yellow sands (dunes); (b) SMU 2: Primarily types 4 and 5 biocrusts on shallow calcareous sandy loam; (c) SMU 3: Types 1-5 biocrusts on deep calcareous sandy loam (Photographs by S. Doudle, 2011)



(b)

Figure 3: (a) Photo taken of different biocrust stages North of Stockpile 19 (near Site 9); (b) from left to right, Bare, Early and
Late stages with biocrust sample from Late stage (excised).





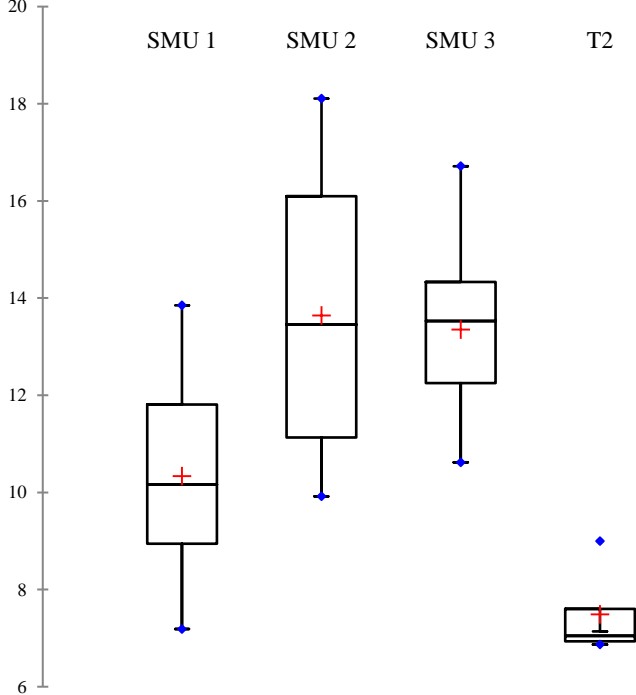

Figure 4: Chlorophyll concentration (µg g$^{-1}$ soil) displaying the minimum, 1st quartile, median (-), mean (+) and 3rd quartile displayed together with both limits (●) beyond which values are considered anomalous following 60 days incubation and resurrection (2 weeks) following desiccation period of 6 months. SMU = Soil Management Units where SMU 1 = Sites 4, 9, 10; SMU 2 = Sites 1, 5, 8; SMU 3 = Sites 2, 3, 7; T2 = 2YO Topsoil stockpile originating from SMU 3.



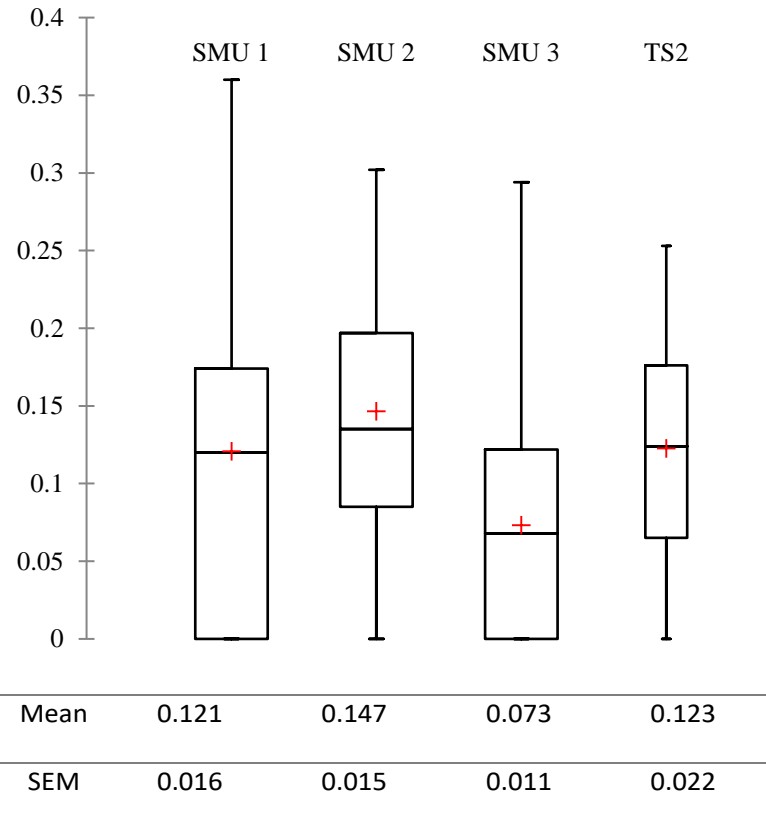

| | SMU 1 | SMU 2 | SMU 3 | TS2 |
|------|-------|-------|-------|-------|
| Mean | 0.121 | 0.147 | 0.073 | 0.123 |
| SEM | 0.016 | 0.015 | 0.011 | 0.022 |

Figure 5: Photosynthetic yield (YII) of photosystem II (PSII) for SMU 1–3 and Site 6 (topsoil stockpile) their mean values and SEM (stand error of the mean) below. The box plots display the minimum, 1st quartile, median (-), mean (+) and 3rd quartile displayed together with both limits.



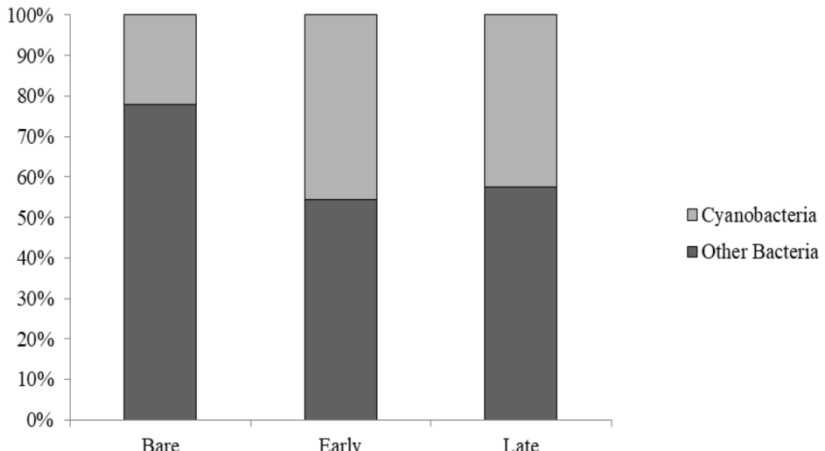

Figure 6: Relative abundance of cyanobacteria to other bacteria within Bare soil and Early and Late stage biocrusts

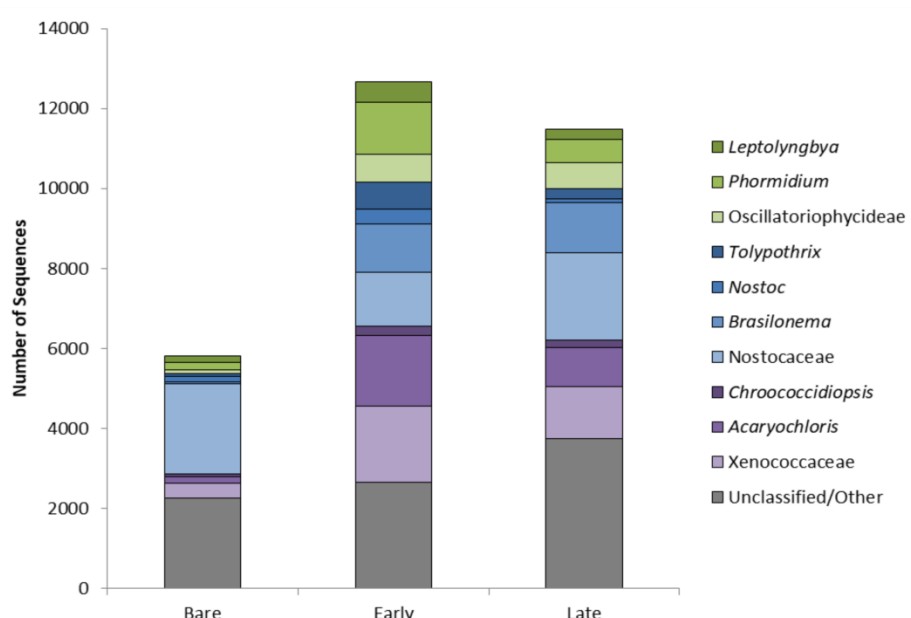

Figure 7: Abundance of cyanobacterial genera and groups. Green = Simple filamentous types, Blue = Heterocystic types, Purple = Unicellular. Unclassified/Other includes chloroplasts.



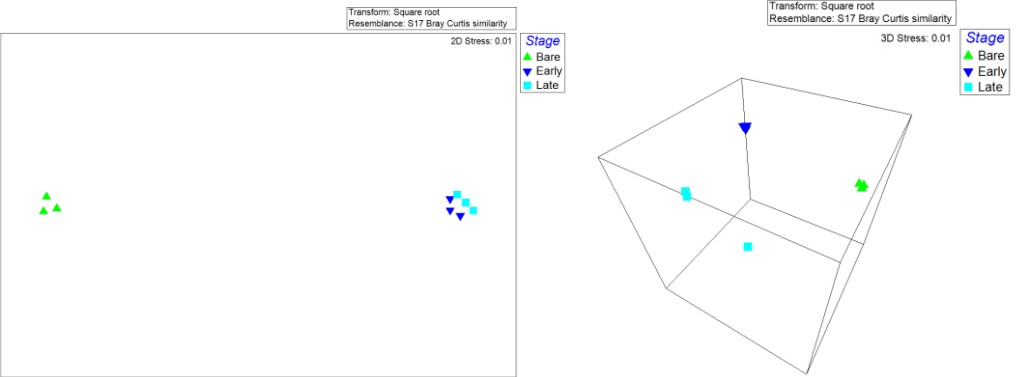

Figure 8: Non-metric Multidimensional Scaling of Bare soil and Early and Late Stage biocrusts within two dimensions (left)
and three dimensions (right).




(a)

(b)

(c)

(d)

(e)

(f)



Figure 9: (a) Predominantly subsurface-dwelling cyanobacteria *Microcoleus paludosus* forming a network of filaments across the soil surface during times of adequate moisture and light; (b) Masses of fine filaments of surface dwelling *Symploca* embedded into EPS that provides a gelatinous and sticky biofilm to create the basis of the biocrust; (c) *Schizothrix* contained in thick sheaths; (d) biocrusts were typically 1-10 mm thick and could be easily removed in pieces from the soil, white arrow points to tiny black cyanobacterial colony with UV protection in amongst green colonies; (e) gelatinous EPS covering cyanobacterial colonies; (f) a probe is used to lift cyanobacterial crust that is held together by surface and subsurface network of filaments.

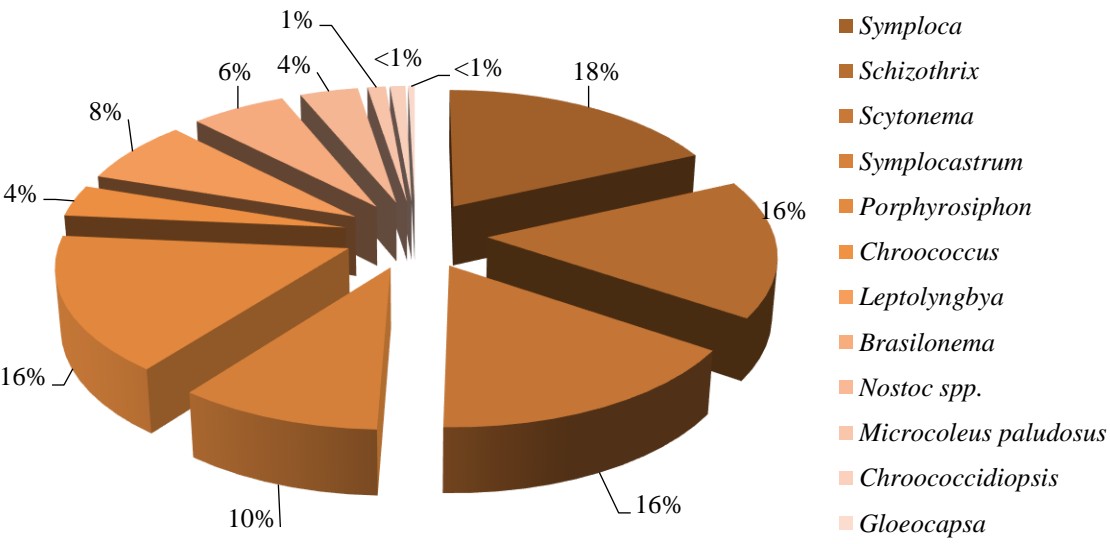

Figure 10: Cyanobacterial community structure across all sites expressed as a percentage of the total community based on mean diversity and abundance scores. N-fixing cyanobacteria contributed to 61% of the community diversity.





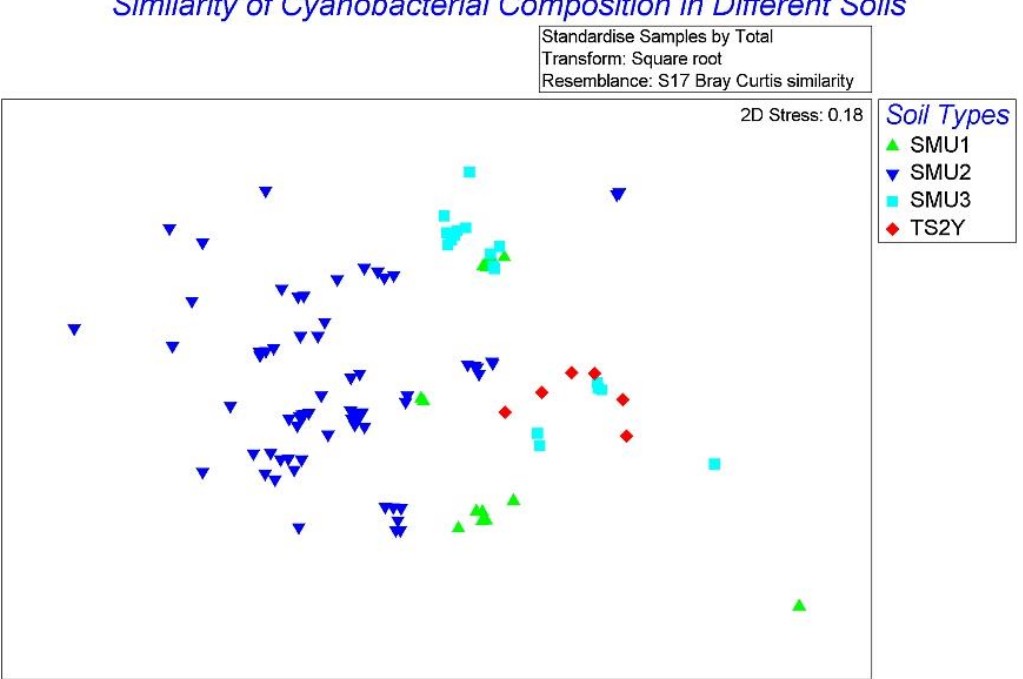

Figure 11: Cyanobacterial community structure based on indexed abundance and diversity across all sites displayed in an nMDS plot (Bray Curtis similarity). SMUs 1-3 refer to key soil management units; SMU 2 appeared to have more range in the clusters compared to the closer groupings of SMU 1 and 3. TS2Y is the 2YO topsoil stockpile; note these are clustered more closely.



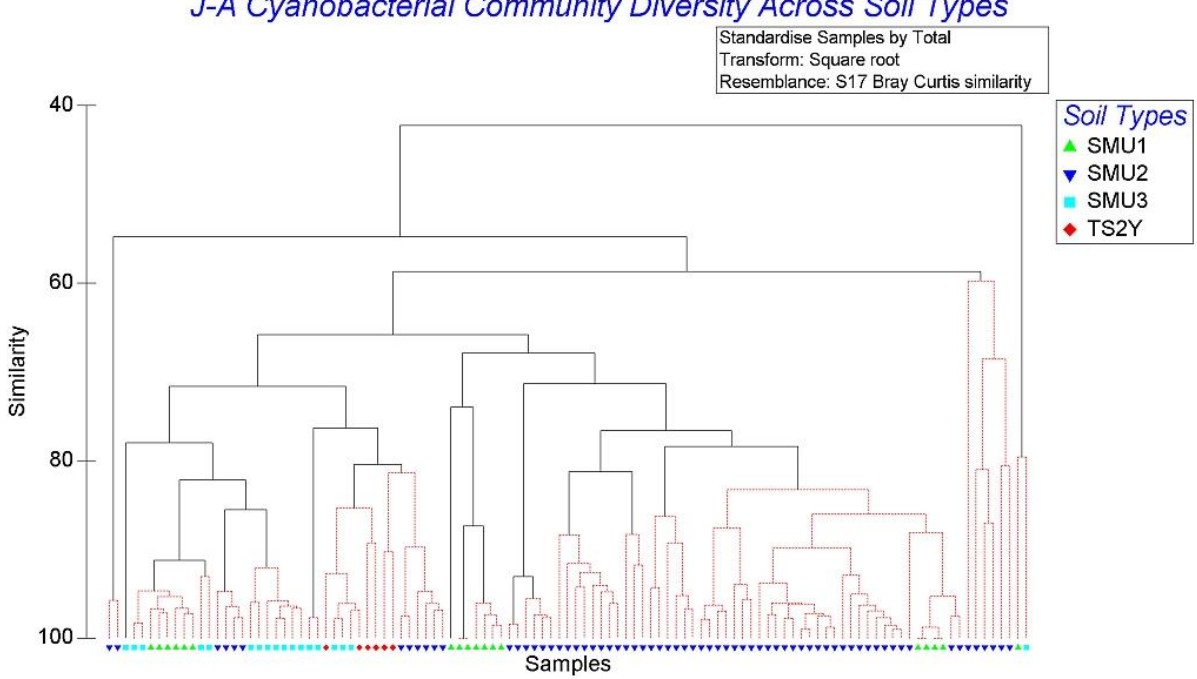

Figure 12: Similarities between samples within their SMU's are displayed in a Bray Curtis dendrogram. Black continuous lines show significant differences between samples (p = 0.05) and lighter lines indicate that most samples were nor significantly different to each other.





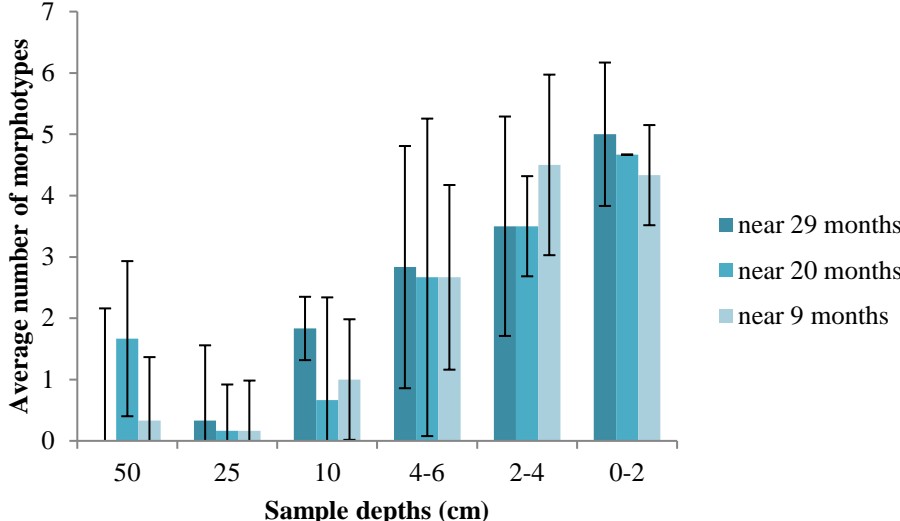

(a)

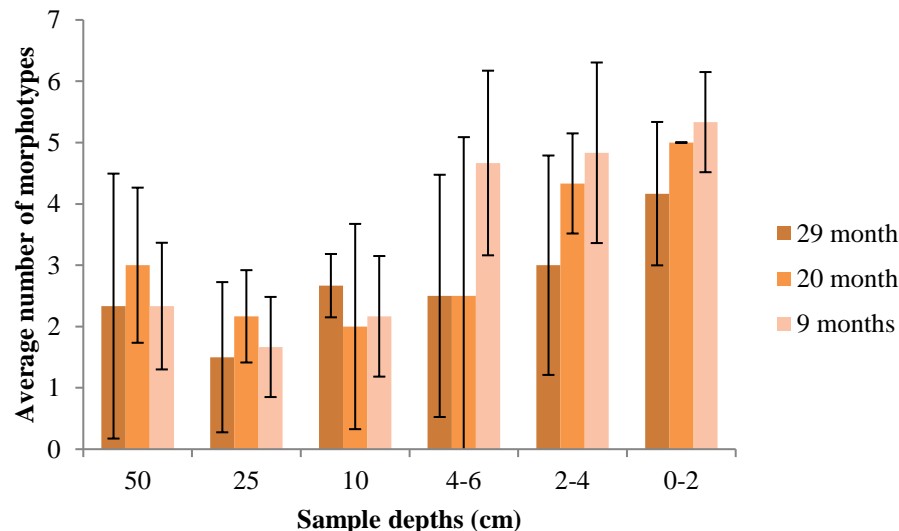

(b)

Figure 13: Cyanobacterial diversity in: (a) undisturbed and, (b) stockpiles of different ages.





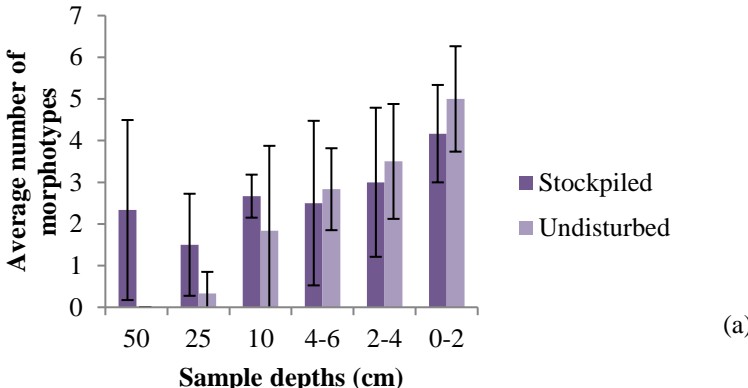

(a)

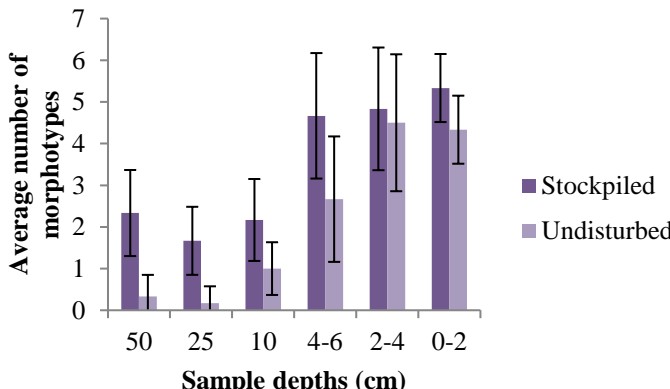

(b)

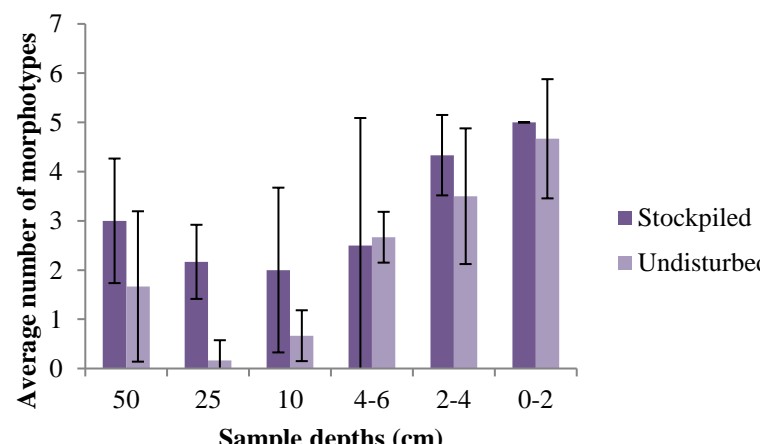

(c)



Figure 14: (a) cyanobacterial diversity following 29 months of stockpiling; (b) cyanobacterial diversity following 20 months of stockpiling; (c) cyanobacterial diversity following 5 months of stockpiling.

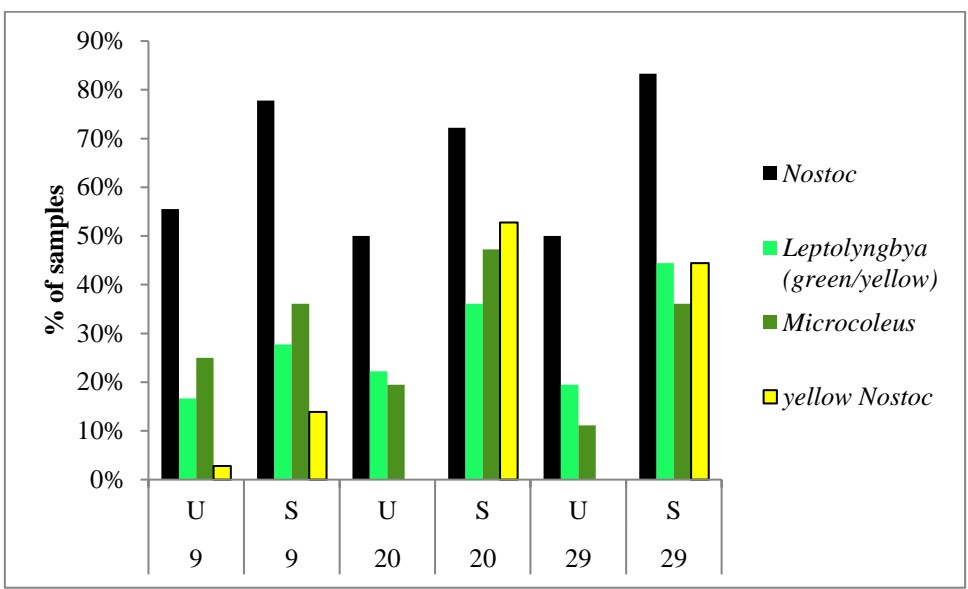

Figure 15: Morphotypes identified in a higher percentage of stockpiled samples (S) when compare with undisturbed samples (U) of varying

5    ages (months)

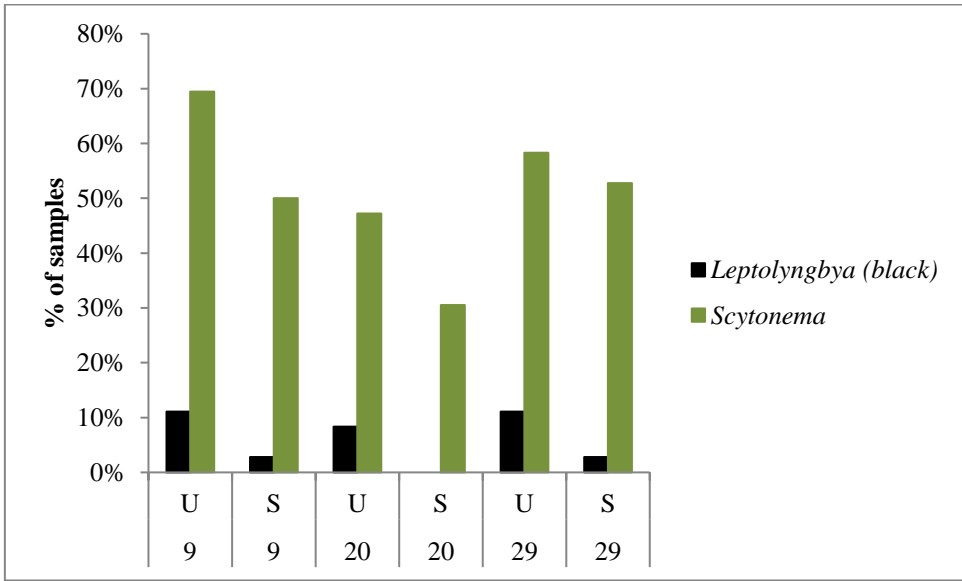

Figure 16: Morphotypes identified in a higher percentage of undisturbed samples (U) when compared to stockpiled samples (S) of varying ages (months)





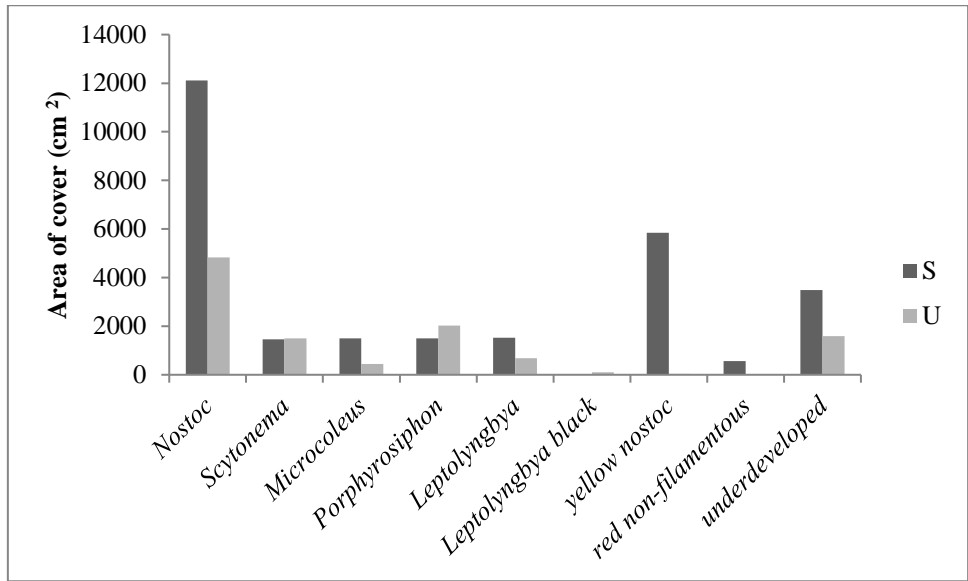

Figure 17: Total area of cover for each cyanobacterial morphotype in all stockpiled (S) and all undisturbed (U) samples





**BIOCRUST RESTORATION: MICRO-PROCESSES**

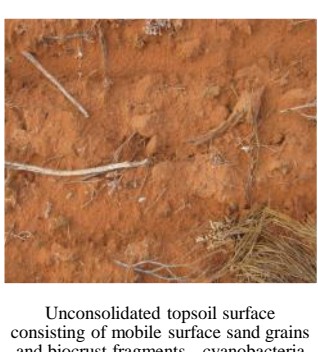

Unconsolidated topsoil surface
consisting of mobile surface sand grains
and biocrust fragments - cyanobacteria
lie dormant until wet

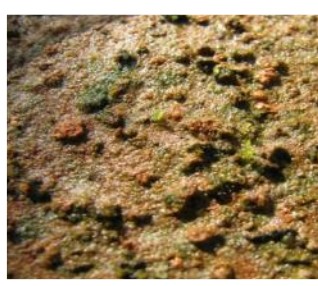

Type 1 Crust formation - increased
chlorophyll content as cyanobacteria
grow and form a smooth surface crust

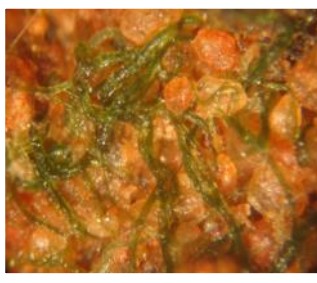

Cohesive strength increases as
filamentous cyanobacteria grow and
subsurface species migrate to surface

Disturbance by wind, raindrop impact and animals can easily
break new biocrust exposing underlying unconsolidated sand

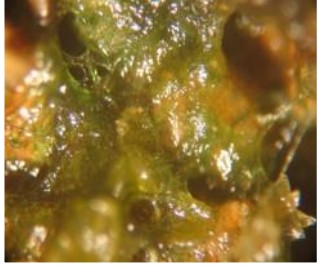

EPS production cements crust, surface
roughness increases and the organic and
inorganic layers form also resulting in
increased compressive strength

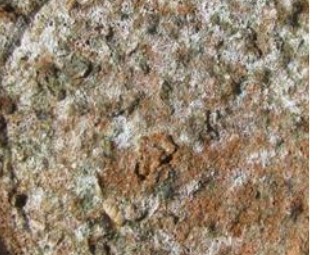

Episodic environmental stress may
result in photo-damage and the
bleaching of EPS; chlorophyll content
declines as some cyanobacteria die

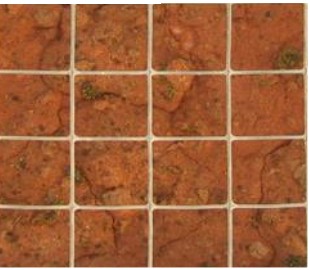

Type 2 Biocrust regrowth following
disturbance incorporating surface
roughness, sand grains trapped in crust
and chlorophyll content increases
(example of photo-grid)

Biocrust re-establishment occurs over time as
seasonal conditions suit and moisture is present

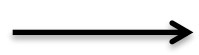

Figure 18: Micro-processes interlinked with biocrust re-establishment, commencing from unconsolidated topsoil to the regrowth of biocrust Types 1 and 2.





## Acknowledgements

This project was funded and supported by Iluka Resources Ltd. and the Jacinth-Ambrosia Rehabilitation team. Thanks to Sam Doudle, Iluka Rehabilitation Specialist for her recognition of the importance of biocrusts at Jacinth-Ambrosia and for initiating this research program. This was a UniQuest project that was carried out at The University of Queensland's School of

Agriculture and Food Sciences, Gatton Campus and the University of New South Wales. Especially appreciated was the strong support, project advice and laboratory resources provided by Assoc. Prof Vic Galea and Katherine Raymont. Special thanks to Dr Glenn McGregor for technical advice, assistance with cyanobacterial identification and the use of specialist microscopes and facilities at the Ecosciences Precinct (Qld), as well as report editing. Special thanks to David Tongway who assisted in Landscape Function Analysis field work, editing and with valuable insights and contributions to this manuscript.

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
