# Peer review of "Microbial Biobanking Cyanobacteria-rich topsoil facilitates mine rehabilitation"

_Biogeosciences, 2017_

## Referee Comment (RC1) · Anonymous Referee #1 · 18 Dec 2017

GENERAL COMMENTS The article title Microbial biobanking cyanobacteria-rich topsoil facilitates mine rehabilitation, by Wendy Williams and collaborators, explores a very important topic in ecological restoration: the rehabilitation of degraded soils provoked by mining activities. And it is relevant because the work was done in a dryland, and drylands are highly impacted by human activities, which are responsible for the continuous loose of ecosystem goods and services provided by healthy environments. The experiment (or the group of experiments) looks like to be well-designed and the methods, poorly explained, are, in general, correctly applied. Actually, the aspect the most I like of the manuscript is the use of both classic and novel methodologies to explore, describe and study the community of microorganisms, mainly cyanobacteria. It is not common at all to find works that use both approaches at the same time. But the problem is that the methods are not well described, so it is really difficult to understand at first sight what authors have exactly done. And because of that, the results are not clear, and everything in this section is a mess, which it is a pity because results are really relevant for the field of degraded soils in arid-lands, and their restoration. Finally, the discussion section could have a great potential and impact in the current literature on the topic, but it seems to me that the way in which authors have organized this section deeply penalizes the whole work. In its current condition, I do not recommend this manuscript for publication. Moreover, I recommend the authors to read the manuscript with attention because they will find some parts difficult to understand: sometimes it is necessary to include punctuation, and some other times they wrote the manuscript fast with little attention to the meaning of the sentence within the paragraph. But, if authors make a great effort and change everything I do not like in the manuscript, I will take a look again with a lot of pleasure.

SPECIFIC COMMENTS (by section) ABSTRACT Although well written in general, I miss more specificity in the section. For example, any general result of the study is provided, so it is difficult for any potential and/or interested reader to decide how to do with the manuscript (whether to continue reading it or not). So, please, provide important results within the section (the community structure that you found in your study, the level of chlorophyll a, etc.). Regarding the first sentence of this section, I think that mining rehabilitation does not require any key solutions, but the degradation provoked by mining activities, so please rewrite this sentence. In lines 21-22, you say that "...a range of attributes that contribute to their resilience and survival in arid environment.": well, this is true, but not only for the organisms living here but for any of them living anywhere, so please remove this idea, which is not relevant for your study, and not only here but in some other parts of the manuscript too.

INTRODUCTION In my opinion, this is the best section of the article. It is well written, it is easy to follow and understand, and both the general problem in rehabilitation programs in arid lands and the specific goals of the current work are provided (although

the specific goals are difficult to understand), all along with good but old references. In this sense, please, provide some other recent references (in this section especially in line 10, page 2). And provide some extra references all along the manuscript too. For example, you have not included any reference of recent works focused on restoration of degraded soil in arid lands with cyanobacteria-based methodologies. You provide some references of rehabilitation works, but most of them related with works that include mosses only. I want to highlight several problems that I see in this section. Please, do not mention biofilms too much, this term is more restricted to aquatic environments, and you work with soils (line 24, page 2). In line 27, page 2, you say chlorophyll, but what type of chlorophyll you measured? I suppose is a, but you don't say anything about it, even in the M&M section, so we need to suppose that you measured chl a. Please, bear in mind that this pigment is the only photosynthetic pigment of cyanobacteria, and this is why you measured it. And you do not say why you measured this pigment. Again, it is necessary to suppose that you measured it as a surrogate for biomass, which is correct, but this is not the only way to estimate biomass in an indirect way. So, as I will say later, be much more specific, you have to provide clear information and data. In line 32, page 2, please replace "was" by "were". In line 9, page 3, you say that "Research into...is rare". Compared to? Please, be more specific here too. In line 22, page 3, replace "in a plant-available" by "in a biological-available". Not only plants will benefit with the presence of N2 fixers in soils of arid lands. What do you mean with the idea of the sentence in lines 24-25, page 3? I really think that this work has only focused on the scale of community of microorganisms, right? But because the methodology is obscure, I am not sure about it. And perhaps I am not able to understand this sentence because you are not clear enough when you write it. In any case, try to read the section again and look for sentences to be improved, there are some of them too long to understand.

METHODS This section is, in general, a mess. And it is a pity because the work that you have done has a lot of potential, as I said before. The section needs to be rewritten again, starting from the beginning. For example, you say that you were working in 10

sites, but in Figure 1 we have 11. It is really difficult now to know how many points, sites, samples, etc. you were working with. Please, imagine that you are a reader instead of an author, and rewrite the section according to it. You have to write in an easy way, in order for any potential reader to understand what you want to say. If I understood the manuscript, you were working with natural crust samples, and also with stockpile samples. But this is not said at the very beginning of the section, and you should write it. Even in the previous section you have to mention that you worked with both kind of samples. In line 26, page 4, what you mean with crust types? It is essential that you define the crust types here, because you use them all along the manuscript. In line 2, page 5, it is not enough to cite this paper. You have to provide clear, concise specifications about the properties of your different biocrust types. In line 11, page 5, what kind of tool you used to excise biocrust and take samples? Again, please, be more specific. In line 3, page 6, again, what kind of chlorophyll you were working with? I suppose that it is a, but you do not say anything about it. Even if you provide a reference that you followed to estimate chl a concentrations, please briefly explain what you did, what kind of devices, if any, you used, etc. Be more specific again. In lines 10-11, page 6, you say that "The measurement was taken at the point when the crust was broken. . .". This does not have any sense to me. You have to measure your variable with the penetrometer in intact soil cores, otherwise your results are not meaningful or representative. When explaining the measurements of photosynthetic activities, please, do not give technical explanations, but provide information on what you did. Now, it is difficult to assess what you have done. I know that new molecular techniques change constantly and you need to be an expert to provide up to date specifications on what you do. But in your case, you have not said enough to understand what you did with your samples to identify microorganisms in your samples. The only point of all the process that you followed which is clear to me is the first one: the extraction method of the DNA of the soil. But after that you have not provided any information. For example, how you built the libraries for amplicon analyses? What kind of steps you followed in the PCR cycles? What kind of technique you used for sequencing the samples? Please, be more

specific here, because this part is, apparently, central for the rest of the manuscript. Because a good point of the manuscript, as I have said before, is the use of classic and novel methodologies to identify cyanobacteria, I would merge both subsections in the M&M section saying that the community structure of cyanobacteria was analyzed by using both approaches and... In any case, it is not clear now if you used both approaches for both type of samples (biocrusts and stockpiles). In my opinion no, but because everything is a mess in this section, any future reader will have the same problems that I have now to understand the methods. Finally, please, specify what kind of statistical analysis was used for each specific question of the work. Because the questions of the work are not provided in a clear way, to specify statistical analyses is challenging in the current form of the manuscript. Please, also specify the significance level you chose, this is really important. Go step by step, now this subsection looks like a salad of statistical tools to address unclear questions.

RESULTS This section has a great potential, but I think that there is lot of room for improvement in the interpretation of the results. Perhaps because the methodology is not appropriately explained, this section is hard to follow, and patterns in results difficult to discern. Please, present your results step by step, use your experiments (or what you want) to do it. Otherwise, it is really difficult to extract the important information that you saw after your excellent experimental design. In line 24, page 8, do not start the section saying "In Table 1 we present...". This is not elegant. In line 25, page 8, what you mean with "...ecologically significant differences..."? Differences can be (or cannot be) significant, and after this you provide an ecological/biological explanation. But not the other way around. In line, page 9, again, specify the type of chlorophyll that you measured in your samples. In the same line, you forgot to put a number of chl concentration, and this means that you did not pay enough attention when you wrote the manuscript, and you did not spend enough time for editing it. This is very serious. Now, I have to say that it is much better to use surface units instead of weight units when pigment concentration data are reported, because it is much easier to compare different places by surface than by soil weight. And this is because the surface is

always the same but the weight of a soil is highly variable depending on its texture. So, please, if possible, express all pigment data by surface. In line 6, page 9, you introduce the site T2, but I think that I haven't seen it before in the manuscript. Please, double-check just in case. And be sure that you describe in the previous section of the sites, places, treatments, etc. in an easy way to follow. In line 7, page 9, do you know what the Levene's test is used for? I do not think so, but it is normally used to test the homogeneity of variances among groups, not for testing the potential differences in the mean of the dependent variable. Please, remove it from here and explain in the previous section what exactly you did to analyze your data. In line 8, page 9, you introduce a result of a Student-t test, but you did not say anything about it in the previous section. So, please, state in the data analyses section what you did, and be precise because this is really important. I am not sure, because it is not really well explained, if you estimated the community structure of cyanobacteria using classic and novel (molecular) techniques in both type of samples (the three stages and the stockpiles). Please, state in the previous section, in a clear way, what type of technique you used for each kind of sample. In line 19, page 9, you say "... and the unicellular genera ...", but what genus or genera. Again, you haven't paid any attention to edit the manuscript, and it looks like that you want to publish it not matter at all. In line 20, page 9, differences in richness, evenness or diversity, according to what? Explain in the previous section what you did, otherwise it is impossible to follow the results here. In lines 24-25, page 9, the sentence is OK, but it does not belong to your manuscript, so please focus on what you have done, nothing else. In lines 26-30, page 9, you just describe what we know about biocrusts. This is not a result at all, so please remove it. And be careful when you say subsurface cyanobacteria, explain that they can survive under these conditions although they normally need light to be alive. In lines 7-8, page 10, you suddenly say that the soil texture can influence the community structure, which is totally true, but the problem is that you do not say anything about it before, so I have to suppose that you measure soil textures but I can't see any result in the manuscript. In lines 8-9, page 10, the ideas are opposite. Which idea is the

correct one? In line 16, page 10, you introduce an idea about the soil textures, please, do it before. In this subsection (3.5) there are too many numbers when you present your data on community structure. Please, sum up this information and only show the real important numbers, otherwise nobody will understand anything. In line 1, page 11, you say "...2YO topsoil...", and although I think that I am able to understand what you mean, it is difficult, and any reader wants to think, just to read and understand your work. In line 13, page 11, you did not sample any Stigonema, you observed it under the microscope. In line 16, page 11, and in other parts of the manuscript, when you show results of data analyses, please indicate what kind of analyses you used, because now it looks like a salad of numbers. And, please, again, in the data analyses subsection of the previous section, state what you did, why you did, etc. In line 1, page 12, area? Why area? You have not measured any variable by area, or you have? Please, be concise and explain what you did. In lines 19-20, page 12, please remove the whole sentence, this information is not really relevant for your work. In lines 21-22, you suddenly start talking about growth rates. So I can suppose that you measured them. But you did not say anything before!!! Please, rewrite the whole M&M section and, after it, rewrite this section again.

DISCUSSION The most important section of any paper is poorly organized, hard to follow, and there is a number of comments where I believe the results are being over-interpreted. So, I recommend rewriting the whole section again. In lines 4-5, page 13, you state that "...species having a range of attributes that contributes to their...". This is OK, it is logical and can be applied to any community in the world. We know it. But you have not measured any of these attributes, and this is not the most important contribution of your work. Actually, this is not a contribution at all. So, please, start this section describing what you have discovered or seen after your excellent field and lab work. In lines 8-9, page 13, again, do not use "biofilm". And instead of "microbe" use "microorganism" all along the manuscript. Do not include any summary of results at the very beginning of this section. Leave it for the next section, the conclusion section, if you want to include this kind of summary. In general, I have the impression that you are

not discussing your discoveries in this section. You are just stating ideas that are well-known for people working with biocrusts. And it is a pity, I have to say again, because you did a hard work and you worked with degraded soils after mining activities in drylands, which is really interesting. So, please, make an effort and rewrite this section, showing first your main discoveries and second using what we know to understand what you saw. In lines 15-16, page 14, you start talking about salinity gradients, but you have not said anything before in the M&M section. And this is a recurrent problem of the manuscript. Please, correct it. In lines 12-13, page 15, what kind of chlorophyll are you talking about? And the presence of chlorophyll is indicative of the presence of an organic layer, but not of an organic layer of EPS. Do you really know what EPS means? Please, make an effort and read more literature on the topic. It is well-known that Microcoleus is the pioneer bacterium in the establishment of biocrusts in North America and Asia. Because this result partially contrasts with what you have seen, it would be great if you include a paragraph discussing why you have seen this difference. You can then talk about biogeography, dispersion, etc., and the paper will have a very interesting ecological perspective. Lines 13-16, page 16, are difficult to understand. In lines 24-25, page 16, you are not discussing your own results anywhere, you just say what we know, but this is not the goal of a discussion section. Again, in the subsection 4.5, I do not see any of your results discussed along with what we already know. You just state a lot of ideas, all of them correct. But this manuscript is not a review manuscript, but an original paper I suppose, so, please, include your main discoveries and, simultaneously, discuss them with what we know, but highlighting what you think is really novel after your excellent work.

CONCLUSION In general, and not only in this section, please include references of cyanobacteria-dominated biocrusts. During the last 10 years, a lot of work has been done on this topic, not only in pristine places but also in degraded soils. In any case, you need to rewrite this section, in its current form is difficult to distinguish between your novel discoveries and the well-known patterns and properties of biocrusts.

TABLES Remove Table 2 and make a plot (barplot) with these data. In Table 4, I do not know which one is the dependent variable of the analysis. So, it is impossible to interpret the table and its results. And this again reflects a big problem of the manuscript: you did not spend enough time double-checking all the info provided and, as a consequence, the manuscript is not consistent and has a lot of mistakes. And this is serious if you pretend to publish it. In Table 5, how did you measure the diversity? Or numbers here are richness data? Please, specify, I do not understand.

FIGURES In Figure 1, how many sites did you survey? 10 or 11? Please, be homogenous all along the manuscript. This is important. In Figure 4, I miss the y-axis name. This is a typical error when you are student, not your case. Same in Figure 5, be serious please. . . Same in Figure 6. . . In Figure 7, transform the plot and make a relative abundance one (like the Figure 6). With the new plot, the box called "unclassified/other" will have less weight. In any case, I think that you haven't done too much effort in identifying all these "other" OTUs, but. . . In Figure 11, use the info that you provide in the caption to discuss the results in the manuscript, this is not the place to do it. In Figure 13, I think that diversity is really richness, right? Please, double-check it. Same in Figure 14. Do you really know the difference between richness and diversity? I now have serious doubts.

FINAL COMMENTS In this manuscript, the introduction is well written, but a bit repetitive sometimes. The rationale for the study of degraded soils after mining activities, and its rehabilitation, is clear. And the combination of classic and modern approaches to study microbial communities of biocrusts is interesting. But the manuscript needs a profound change, starting with the methods and finishing with the conclusion, and also including the figures and the tables. Authors need to improve the quality of the manuscript because they did a great job in a current field of work. It would be a pity if they do not do it because we will not have access to important ideas and discoveries on restoration processes mediated by biocrusts in degraded soils of drylands after mining activities.

---

## Short Comment (SC1) · 19 Dec 2017

W. Williams

wendy.williams@uq.edu.au

I would personally like to thank the reviewer for their great amount of time and effort put into providing a clear commentary with respect to improvements needed in this ms. I am confident that these issues can be addressed in a timely manner. I understand the comments and time taken to provide such a substantial review. In an effort to reduce the length of the ms we have clearly left out important information as well as made it appear disjointed. In particular because this ms is a compilation of three key studies carried out by co-authors I recognise the fact that they have not been presented in a nice tightly knit sequence and will put a lot more effort in doing so. Once again thank you for your time and have a safe and happy Christmas. Kind regards, Wendy

---

## Referee Comment (RC2) · Anonymous Referee #2 · 21 Jan 2018

I found the manuscript interesting and encompassing almost all the indicators relating to the populations of cyanobacteria in the area undergoing rehabilitation after mining.

It was also worth adding an illustration of the distribution of rainfall and temperature. I am certain that this data is available and can be added to the manuscript.

The only small comment is that there is no reference concerning the characteristics and major elements in the tested soil types. What are the major elements that can be found and their levels in the topsoil, restored areas and the natural area nearby? From my experience in comparison of the natural soil near disturbed areas, there is a sharp change in the content of the various elements and their dispersion both on the soil surface - topsoil and in the depth of the soil [a few cm]. This change may enhance or suppress the cyanobacterial growth.

---

## Author Comment (AC1) · 12 Feb 2018

Responses to Reviewer 1 We would like to thank Reviewer 1 for their recognition of the importance of this research. The suggestions provided by this thorough review has helped us improve the manuscript considerably. One of the difficulties in the preparation of the ms was the compilation of the work and the transformation of a mine report to a scientific paper. This explains a lot of generalisations especially in the discussion. I believe we have now addressed all your concerns in full. Comments and questions below: R1: Abstract needs more specificity; alter first sentence with focus on degradation; remove sentence from lines 21, 22 as it is not relevant to this study. Response: Abstract fully revised to include specific results and sentence removed. "Abstract Degradation from mining activities requires key solutions to complex issues

where the removal or disturbance of topsoil incorporating soil microbial communities can result in a shift in ecosystem function. The research was in collaboration with Iluka Resources at Jacinth-Ambrosia (J-A) mineral sand mine located in a semi-arid chenopod shrubland in southern Australia. At J-A diverse biocrusts included a significant representation of cyanobacteria, lichens and mosses that inhabited nearly half of all soil surfaces. Cyanobacteria often dominate dryland soils and work as ecosystem engineers, in that they are in sufficiently large quantities to initiate biocrust establishment and facilitate soil surface stabilisation. This research encompassed soil microbial community profiling (using a polyphasic approach) with a focus on 'biobanking' topsoil for rehabilitation purposes. Biocrust successional stages were linked to soil types and formed the basis of the experimental design. Sequencing showed cyanobacteria were a significant component of all three successional stages. Microscopically, 21 cyanobacterial species were identified across the ten sites. Known nitrogen-fixing cyanobacteria Symploca, Scytonema, Porphyrosiphon, Brasilonema, Nostoc and Gloeocapsa comprised more than 50% of the diversity at each site and formed 61% of the total community diversity. There was no significant difference in cyanobacterial community structure across soil types which suggests that diversity and abundance is not controlled by soil type. Chlorophyll a concentrations sourced from the 2-year old topsoil stockpile was 7.49 $\mu$g g-1 soil, almost half the concentration of its source soil (13.53 $\mu$g g-1 soil). A total of nine cyanobacterial morphotypes were identified from the samples from nine stockpiles. Average morphotype richness was highest in stockpiled samples at and above 10 cm depth for all stockpile ages. Biocrust re-establishment during mining rehabilitation relies on the role of cyanobacteria as a means of early soil stabilisation. Ongoing monitoring of biocrust recovery is important as it provides an effective means of measuring important soil restoration processes." R1: Additional up-to-date references required in introduction, in particular p2, L10, include additional cyanobacterial restoration references Response: More recent references have been added to introduction (see edited ms below) and additional cyanobacterial restoration references have been added throughout. Paragraph incorporating P2, L10 now

reads: "Mine rehabilitation is a complex process that involves many levels of under-standing of difficult issues relating to ecosystem function where the removal or burial of the bioactive soils can have knock-on effects for rehabilitation efforts such as na-tive seedling establishment (Jasper, 2007; Tongway and Ludwig, 1996). Successful ecological restoration of arid mining sites relies on a holistic approach where biocrust recovery to pre-disturbance levels is integral and can serve as an indicator of the in-tegrity of the ecosystem (Tongway, 1990). Research into biocrust disturbance with a focus on recovery post-mining is rare. In the Namaqualand arid lands (Namibia, South Africa) low rainfall and high winds impact the rehabilitation of degraded lands follow-ing diamond mining and grazing (Carrick and Krüger, 2007). These researchers found that cyanobacteria and non-vascular plants that form a living and protective surface crust were crucial to surface stabilisation. Jasper, (2007) also recognised the impor-tance of soil microbial communities including cyanobacteria in post-mine rehabilitation in the Jarrah forests of south-western Australia. In the Czech Republic and Germany chrono-sequential studies of old brown coal mine sites found in younger sites green algal biofilms and a diverse range of cyanobacteria initiated the rehabilitation of the soils (Lukešová, 2001). In serpentinite mine tailings (New South Wales, Australia), McCutcheon et al., (2016) showed filamentous cyanobacteria accelerated carbonate mineral precipitation and stabilised the tailings. They demonstrated cyanobacteria had the capacity to adsorb magnesium while acting as a nucleation site and sequestered carbon. In our current study preliminary research identified that in the chenopod shrub-lands at the edge of the Nullarbor Plain (South Australia) biocrusts cover the soil sur-faces between the grass plants and post-mining rehabilitation needs to investigate their role (Doudle et al., 2011). It follows that there is a real need for a focus on practical approaches that contribute to the restoration of soil function and measure relevant aspects of success through soil microbial communities and biocrust reestablishment, especially cyanobacteria (for example: Setyawan et al., 2016; Mazor et al., 1996; Fis-cher et al., 2014; Chiquoine et al., 2016; Doherty et al., 2015; Harris, 2003; Tongway and Hindley, 2004; Zhao et al., 2014)." R1: Use of word biofilm Response: I would

partially disagree with R1 regarding the use of biofilm being more a less restricted to aquatic and marine environments. For example: Rossi and De Philippis (2015) specifically refer to the role of EPS in the creation of a biofilm in arid environments as the first step in biocrust establishment. I have revised its use (and context) and referenced it appropriately. Sentence incorporating biofilm in introduction now reads: "Cyanobacterial biofilms provide initial stabilisation of disturbed surfaces that pave the way for diverse microbial communities, and form bioactive crust-like layers integrated into the soil surface (e.g. Büdel et al., 2009; Rossi et al., 2017; Bowker et al., 2014). R1: define chlorophyll (type) measured (p2, L27) also in M&M section Response: Sentence now reads "EPS more than doubles the biocrusts compressive strength and increases cohesiveness by up to six times with a ratio of at least 2:1, EPS to chlorophyll a (Hu et al., 2002)." Response: see M&M revisions – chlorophyll a has been defined in all areas. In M&M main sentence now reads: "In order to define the cyanobacterial component of the biocrust, chlorophyll a pigmentation (unique to cyanobacteria) was measured following resurrection." R1: p2, L32 change 'was to were' Response: completed R1: "Research into...is rare" be more specific. Response: Refer to paragraph revised paragraph above that has included additional references and a more complete description of known research to date. R1: p3, Line 22 change plant-available to biological-available Response: completed R1: Clarify focus of work Response: We have removed site descriptions and background to start of methods and clarified the lead in paragraph to hypotheses to read as follows: "This project is based on designing mining rehabilitation plans that will achieve improved long-term outcomes. The restoration of landscape function and the accompanying need for the restoration of the soil ecosystem that included biocrusts after high-levels of disturbance directed the development of this biocrust research project. As cyanobacterial communities often develop their species richness, abundance and structure in response to their environment (e.g. Aboal et al., 2016; Büdel et al., 2009; Williams et al., 2014; Williams and Büdel, 2012), it was important to examine the biocrust community structure and survival. Within this design a polyphasic approach considers the essential ecosystem services provided through the

reestablishment of biocrusts. The conceptual design is based on the net ecosystem benefits that must be achieved through biocrust regeneration. It follows that cyanobacterial inoculum in the topsoil stockpiles would be central to early stabilisation of mobile surfaces subjected to the potential impacts of wind and rain splash erosion. We sought to determine whether shallow 'biobanks' of cyanobacterial-enriched top soil would facilitate biocrust recovery of this mine site." R1: Methods need revision for clarity, specific points to address are addressed in responses below Responses: The methods have been completely revised into a clear sequence under revised headings as follows: 2.0 Methods 2.1 Background and site description 2.2 Field Methods 2.3 Ecophysiological properties of biocrust cyanobacteria 2.4 Cyanobacterial community structure 2.5 16S rDNA profiling of native undisturbed biocrust microbiomes 2.6 Cyanobacterial tolerance to stockpiling Specific points addressed below: R1: 'Fig 1 (11 sites) vs. 10 sites sampled' Response: Figure description now states that Site 11 was not used. R1: "natural samples vs. stockpile samples" Response: In the last paragraph of the introduction we had already clearly stated they were natural undisturbed samples versus stockpile samples that had been crushed and buried. "The overall goals of the biocrust research program were to: (a) evaluate specific roles of natural, undisturbed biocrusts in ecosystem function at the mine site; (b) determine cyanobacterial community structure in terms of key species that drive early colonisation, biogeochemical cycling and soil stabilisation; and (c), to investigate the effects of stockpiling topsoil on cyanobacterial survival after burial and subsequent recovery." R1: 'Explain crust types' Response: We have added a description into the first section 'Site description and background' "The biocrusts at J-A had been previously classified into three primary successional stages representative of the five biocrust types found growing across the landscape (Doudle et al., 2011); Types 1–2: light coloured, thin cyanobacterial crust in early stages of development; Type 3: cyanobacterial crust, well established, intermediate stages of development; Types 4–5: biocrust, well established with cyanolichens and/or green algal lichens and mosses, late successional stage of development (additional descriptions available in supplementary Table S1). Study locations were selected

from the main vegetation associations across the three soil types and Lake Ifould (a dry salt lake)." R1: 'p5, line 2 what type of tool?' Response: Inserted "Within each area eight 10 cm diameter samples were selected at random and removed to a depth of 1 cm using metal scraper (n=80), air dried (>40°C), and stored in Petri dishes." R1: 'p7, line 3 what type of chlorophyll and improve description' Response: Chlorophyll a defined throughout. Description of methods improved: "Chlorophyll a concentration of the biocrusts were determined following resurrection (by moistening) using a 1:5 ratio of (dry weight) biocrust to Dimethyl sulfoxide (DMSO) (Barnes et al., 1992) with samples placed in a warm bath (65°C) for a two-hour dark extraction, followed by centrifuging for five minutes (5000 g RCF). Chlorophyll a concentration was calculated using Wellburn's (1994) equations." R1: 'p6, lines 10-11 clearly explain use of penetrometer' Response: Description revised to read "A pocket penetrometer (8 mm foot) was used to determine the compressive strength (kg cm-2) of the dry intact biocrusts. Four measurements were taken from each sample location, providing 12 replicates per site. The measurement was taken at the point when the crust was broken, and the foot penetrated the crust surface." R1: 'Explain measurements of photosynthetic activity from what you did' Response: This has been revised to read "Photosynthetic performance (recorded as yield, YII) was measured using pulse-amplitude modulated (PAM) fluorometer (Pocket PAM; Gademann Instruments, Germany). The aim was to demonstrate photosynthetic yield (YII) indicative of active growth of the biocrusts, using the detection of chlorophyll fluorescence from photosystem II (PSII). The sensor was placed onto the biocrust and once started, a series of short pulses of excitation light at high intensity that is amplified resulting in a brief closure of PSII and the measurement of fluorescence yield based on the Genty parameter which is the quantum yield (YII) of the charge separation of PSII (Genty et al., 1989) and recorded on a scale of 0–1 for all photosynthesis. Allowing a short space of time between readings, this process was completed several times for each sample." R1: 'More detail in sequencing methods and was sequencing done for stockpile samples?' Response: Please note that sequencing was not used in stockpiles as unfortunately there was insufficient budget to cover

this. Now the methods are rewritten this is clearer. Page 5, Line 11: Biocrusts were collected using a paint scraper and spatula which were wiped between each sample using 70% ethanol. Figure 3 shows how biocrusts were selected. More information on DNA library generation: Sentence updated to - Molecular libraries of the 16S rDNA V123 hypervariable region generated via PCR as per Chilton et al., (2017) and submitted to the Ramaciotti Centre for Genomics (UNSW, Australia) for a 2x300 bp sequencing run on an Illumina MiSeq instrument. Clearer statistical analyses: Methods section has been re-structured. Each approach now has the specific statistical analyses used under that section. This section in the methods has been rewritten as follows: "2.3 16S rDNA profiling of native undisturbed biocrust microbiomes For genomic profiling of naturally occurring successional biocrust communities, a location adjacent to Site 9 was visually determined to contain Bare, Early (Types 1-2) or Late (Types 3-5) stages of development (Table S2). Biocrust successional features were determined by morphological attributes of pigmentation, thickness and surface roughness as well as the presence/absence of lichens and mosses (Fig. 3), (Chilton et al., 2017). Bare stage was defined by loose soil particles with no biocrust structure. Samples were collected in July 2014. For each successional stage, three replicates were collected that were representative of SMUs 1–3 where a 10 cm2 plot with 95% coverage of the desired biocrust stage was excised to the depth of the crust and non-aggregated soil discarded (Fig. 4). Samples were processed at UNSW, Sydney. Each biocrust replicate for Bare, Early and Late stages of development were homogenised and genomic DNA extraction performed using the FASTDNA Spin Kit for Soil (MP Bio Laboratories, USA) according to the manufacturer's instructions. Molecular libraries of the 16S rDNA V123 hypervariable region generated via PCR as per Chilton et al., (2017) and submitted to the Ramaciotti Centre for Genomics (UNSW, Australia) for a 2x300 bp sequencing run on an Illumina MiSeq instrument. Sequencing data was processed using Mothur version 1.34.0 (Schloss et al 2009) and described in detail in Chilton et al., (2017). Singleton and doubleton OTUs were removed and samples rarefied to 8598 sequences each across 3785 OTUS. The curated Greengenes database (McDonald et al 2012) was

used to assign taxonomy to OTUs. Diversity values were derived using the DIVERSE function within the Primer package (Anderson et al 2008) upon standardized OTU values. ANOVA with post hoc Tukey's tests was used to test for significant differences between stages. Multivariate analyses were performed in Primer upon a Bray-Curtis dissimilarity matrix generated from square-root transformed abundance data. Samples were represented in two and three-dimensional space within a non-metric multidimensional scaling plot (nMDS). Pair-wise, a posteriori comparisons of factor Stage were performed using the PERMANOVA function with 9999 Monte Carlo permutations. Homogeneity of dispersion for each stage was tested using PERMDISP." R1: 'Please provide the type of statistical analysis for each specific section' Response: see above and further descriptions added throughout revised methods (refer to main ms). Results R1: Please present your results step by step Response: The results section has been revised to reflect the methods section, main headings as follows: 3.0 Results 3.1 Ecophysiological properties of biocrust cyanobacteria 3.2 Cyanobacterial community structure 3.3 16S rDNA profiling of native undisturbed biocrust microbiomes 3.4 Cyanobacterial tolerance to stockpiling

R1: p8, line 24 recast sentence "in Table 1..." Response: this sentence has been removed as it was unnecessary. R1: p8, line 25 'what do you mean by ecologically significant...' Response: the word "ecologically" has been removed R1: p9 define chlorophyll type, add mean concentration (missing), surface area reporting preferred Response: chlorophyll a inserted, mean concentration added. In this case we reported in $\mu$g g-1 soil as we needed to compare with disturbed topsoil and topsoil stock piles. It was later used (ms in preparation) to define the concentrations per g soil to add in restoration trials. I understand that globally comparisons by surface area are easier however we were constrained by the requirements of the mine project. We do however have some earlier biomass area data done as part of an honours project that was carried out as a preliminary study that I will add to final ms revisions. R1: p9, line 6 – had T2 been introduced previously? Response: T2 had been defined in two figure descriptions "T2 = 2YO Topsoil stockpile originating from SMU 3" however in the script

two-year has also been described as 2YO stockpile (see below) and further down has had (T2) added for clarity of reference between results and figures. R1: clarify sites in this section Response: The first sentence now reads "Soil pH across the three soil management units (SMUs) ranged from 8.4–8.6 while the two-year old (2YO) topsoil stockpile was higher at 8.9 (Table 1)." R1: p9, line 7 incorrect use of Levene's test, explain analysis Response: this was removed and correctly identified as not relevant, analysis described in revised methods R1: please state clearly what methods were used for each section i.e. community structure, stockpiles, were polyphasic approach used across all of these? Response: This has been clarified in revised methods and explained in previous section (sequencing was not used in stockpiles as unfortunately there was insufficient budget). R1: p9, line 19 'unicellular…' what genera does this refer to? Response: inserted "(e.g. Chroococcidiopsis, Acaryochloris, Xenococcaceae)." These are also identified in Figure 7. R1 p9, line 20 – explain differences in richness, evenness and diversity according to…. The methods section has been re-structured to better reflect which statistical analyses were used for which set of data. Please find the relevant sentence: "Diversity values were derived using the DIVERSE function within the Primer package (Anderson et al 2008) upon standardized OTU values. ANOVA with post hoc Tukey's tests was used to test for significant differences between stages." R1: p9, line 24-25 also 26-30 remove sentences, not relevant to results or stating what we know. Response: sentences removed R1: p10, lines 7-8 soil texture can influence community structure…not in results? Response: The results relate to soil types rather than textures and written as soil types (SMU 1-3, soil management units) which were previously determined by a soil consultant and described in detail in methods section 2.1 (see below) R1: p10, lines 8-9 ideas are opposite? Response: this sentence clarified R1: p10, line 16 'introduce soil textures, please do it before…' Response: Soil management units are now fully described in Methods Section 2.1 "The landscape has been characterised into three distinct soil types associated with vegetation communities (Table 1) called soil management units: SMU1 – deep calcareous yellow sands associated with dune ridges; SMU2 – shallow calcareous sandy loams and, SMU3 –

deep calcareous sandy loam (Goode, 2009; Doudle et al., 2011)." "The landscape has been characterised into three distinct soil types associated with vegetation communities (Table 1) called soil management units: SMU1 – deep calcareous yellow sands associated with dune ridges; SMU2 – shallow calcareous sandy loams and, SMU3 – deep calcareous sandy loam (Goode, 2009; Doudle et al., 2011)." R1: Section 3.5 'too many numbers in community structure data' Response: This section has been refined as follows: Of the 21 species more than half (12 species) of cyanobacteria were identified in SMU 1 where four primary genera made up 75% of the community: Symploca, Schizothrix, Scytonema and Symplocastrum (for more detail see Fig. S4). Cyanobacterial crusts from the dune regions on SMU 1 (deep calcareous yellow sands) were representative of crust types 1–3; patchy, brittle (when dry) early-successional crusts as well as formed dark crusts that were mid to late-successional and included cyano-lichens (also see Doudle et al., 2011).

Cyanobacterial crusts from the chenopod shrublands and open woodlands in SMU 2 (shallow calcareous sandy loam) represented a broad range of crust types (2–5) but overall could be described as late-successional. Lichens and mosses were highly visible (also see Doudle et al., 2011). There were 21 cyanobacteria recorded: four were primary genera that made up 63% of the community including: Schizothrix; Porphyrosiphon; Scytonema and Symploca (for more detail see Fig. S5). Cyanobacterial crusts from the open woodlands in SMU 3 (deep calcareous sandy loam, Fig. 2c) represented a broad range of crust types (2–5) but like SMU 2 could also be described as late-successional. Lichens and mosses were highly visible (see Doudle et al., 2011). There were nine cyanobacteria recorded of which four were primary genera that made up 85% of the community: Symploca, Porphyrosiphon, Scytonema and Schizothrix (for more detail see Fig S6). Cyanobacteria with the capacity to fix nitrogen contributed to 77% of the community structure. Cyanobacterial crusts from Site 6 were from the 2YO topsoil stockpile that had originated from SMU 3 (deep calcareous sandy loam) would be described as early successional crusts with some seasonal mosses. There were eight cyanobacteria recorded of which four were primary genera that made up

84% of the community: Symploca; Symplocastrum; Porphyrosiphon; and Scytonema (also see Fig. S7). It was interesting to note that Symplocastrum was co-dominant with Symploca whereas in the other communities it ranged between 8-13%. Sub-surface species Schizothrix (found in top 5 mm) only contributed to 4% of the richness compared to 10-20% elsewhere. Cyanobacteria with the capacity to fix nitrogen (Symploca, Porphyrosiphon, Scytonema and Brasilonema) contributed to 61% of the community. R1: p11, line 1 2YO topsoil acronym query Response: this has been addressed early in results (see above) R1: p11, line 13 'you did not sample Stigonema...' Response: This paragraph has been recast to read: "Examination of stockpile soil samples via microscopy revealed five cyanobacterial morphotypes corresponding with the genera: Nostoc, Scytonema, Microcoleus, Porphyrosiphon and Leptolyngbya (Fig. 17). Average morphotype richness was highest in stockpiled samples at and above 10 cm depth for all stockpile ages (Figs. 13, 14)." R1: p12, line 1 'area mentioned....where has it been introduced previously.' Response: In methods area has been described as 'area of coverage' – "To determine cyanobacterial growth rates and richness, wetmounts for each sample were examined under 16 x magnification for cyanobacterial thalli and colony size was estimated via area of coverage of the field of view." Subsequently, the references to the word area in results were changed to cover or coverage to better reflect the description in the methods. R1: p12, lines 19-20 remove sentence Response: Sentence removed R1: p12, lines 21-22 use of the term growth rates Response: above mentioned sentence highlighted refers to growth that was measured over time (via area of coverage). Discussion R1: Please rewrite Response: The discussion has been rewritten with the useful comments by R1 incorporated. The full discussion is added below: 4.0 
[revised manuscript text omitted]
 reactive to cyanobacterial regeneration. As even low-profile stockpiles are in the vicinity of 2 m additional time would be needed for re-establishment of biocrusts which would likely be dependent on rainfall. It follows that the timing of rehabilitation would be important so as to take advantage of favourable climatic conditions. Cyanobacteria are well adapted to long periods without water, the optimisation of short growing seasons, wet-dry cycles, low water potentials, tolerance of high UV and low light intensities, fluctuating temperatures and in some cases high salinity. Cyanobacterial strategies central to survival include EPS production, spectral adaptation, nitrogen fixation and motility. Biocrust re-establishment during mining rehabilitation relies on the role of cyanobacteria as a means of early soil stabilisation. Provided there is adequate cyanobacterial inoculum in the topsoil stockpiles their growth and the subsequent crust formation should take place largely unassisted. Ongoing monitoring of biocrust recovery is important as it provides an effective means of measuring important soil restoration processes. Detecting increases in key species and shifts of community structure will likely provide more informative and robust verification of desired rehabilitation outcomes. Cyanobacterial species richness is an important measure of biocrusts that incorporate microprocesses central to a healthy and functional soil ecosystem. Increased cyanobacterial biomass is likely to also be a good indicator and reliable metric. Diversity indices derived from sequencing data of the whole bacterial community are poor measures of biocrust formation and development.

Tables and Figures: All corections have been made including: Table 4: Table heading

updated to include dependent variable "Table 4: Permutational analysis of variance (PERMANOVA) of pair-wise comparisons of Bray-Curtis dissimilarity between biocrust stages and bare soil" Figure 6: Axis added Figure 7: Graph transformed to relative abundance
* * *

---

## Author Comment (AC2) · 12 Feb 2018

We would like to thank Reviewer 2 for their appreciation of this manuscript and helpful comments to improve its quality. R2: It would be worth adding information on the distribution of temperature and rainfall Response: We have now included the Bureau of Meteorology graphs for temperature, rainfall and evaporation (as this is extreme) for Nullarbor, the nearest recording station with records commencing (with some short breaks in data) from 1888. In case the editor would prefer it as supplementary material we have added a brief climatic description in site background. R2: No reference concerning the major soil types and surface as they could influence rehabilitation of biocrusts Response: We have now included a key reference to the history and nature of the soils from J-A (Hou and Warland, 2005) and incorporated a section in the

revised discussion that focuses on the role of the soil elements investigated in rehabilitation especially in terms of biocrusts. It should be noted at the stage of our research that the first rehabilitation site had not commenced thus we were investigating the natural soil environment of the biocrusts (undisturbed) compared to stockpiles of topsoil which were due to be used in the near future (and subsequently a few hectares have undergone rehabilitation).

---

## Author Comment (AC3) · 12 Feb 2018

Note to editor This ms has been completely revised taking into account the comments from R1 and R2. In summary: The abstract now includes specific reference to results. The introduction is more concise with up to date references included. Methods have been entirely rewritten. Results - all problems with tables and figures are addressed. Discussion has been more sharply focused to results. Thank you for the opportunity to respond to these issues and I believe the ms has been considerably improved.

---

## Author Response (AR2)

We would again like to thank Reviewer 1 for taking the time to carefully read and comment on our manuscript. We have addressed these comments and suggestions below:

**Abstract –**

5  Page 1 Line 10: Change has been made

Page 1 Line 14: Change has been made

Introduction –

First sentence: Change has been made

Page 2 Line 5: Change has been made

10  Page 2 Line 6-7: That section has been removed

Page 2 Line 9-10: The section has been relocated

Biological soil crust is introduced first, with biocrust in brackets afterwards

Page 2 Line 14: Change has been made

The third paragraph has been restructured

15  Page 3 Line 26: Change has been made

Overall: Introduction restructured for clarity

**Methods –**

After Figure S1: www. has been added

Field v Lab: Section made clearer

20  Page 6 Lines 7-11 now clearly explains method: 'A pocket penetrometer (8 mm foot) was used to determine the compressive strength (kg cm$^2$) of the dry intact biocrusts samples. Overall the crust thickness was < 0.5 cm. Each sample (10 cm diameter x 2 cm depth) was placed on a solid surface and a total of twelve measurements (3 readings x 4 reps) were taken for each site. The measurement was taken at the point when the crust was broken, and the foot penetrated the crust surface.'

Page 7 Lines 9-10: Section re-written for clarity

25  Statistical analyses: We initially agreed with the reviewer regarding having a separate data analysis section outlining all analysis performed. However, we found that it was difficult to attribute each statistical method to the relevant task without becoming repetitive and challenging to follow. We found having the statistical analysis in the same section as the task allows for the reader to understand and evaluate each section separately, making it easier to follow the overall aims and story of the paper. The p value is added to each result/figure where required.

30  **Results –**

The "2YO" phrasing has been removed and terminology made consistent throughout

Page 10 Line 3: Change made

Page 10 Second paragraph: This section has been re-written for clarity

Discussion of co-occurring nitrogen-fixing species and their role as pioneers: We thank the reviewer for highlighting these findings for us. We have added a section to the discussion to explore the implications of finding these species and their co-occurrence in early stage biocrusts.

Page 11 Lines 10-11: Indeed, the DIVERSE function within PRIMER was used to calculate the diversity values here. This method is mentioned in the corresponding methods section for the genomic profiling. While we have trialed a standalone statistical analyses section, we found it complicated the flow of the manuscript.

**Discussion –**

Page 13 Line 7: comma added

Page 13 second paragraph: we have rewritten this paragraph for clarity

Finally, we appreciate the reviewers' comments regarding the difference species involved in crust building (i.e. what we have found in Australia vs. common reports elsewhere) and have added some relevant comments to Section 4.1 (paragraphs 1 and 3).

In response to the reviewers suggestion we also relocated some figures to supplementary material and removed some that were not really necessary.

[revised manuscript text omitted]

---

## Author Response (AR3)

Dear Editor,

Thank you for your revision and suggestions for our manuscript. They have all been addressed and we look forward to your acceptance.

Editor: After the third revision of your work I found that you successfully solved most of the problem pointed by the reviewers, however still some few things that needs to be solved in order to improve overall quality of the paper.

As pointed by reviewer 1 during the last revision process, the indirection has been improved when compared with the previous version, but I still quite repetitive.

1. I think that you can improve it by following the structure suggested by the reviewer:

a. The problem of mine activities on soil degradation in drylands,

b. Potential solutions to this problem by using biological soil crust natural activities,

c. Properties and services provided by biological soil crusts in drylands,

d. Recommendations to use biocrusts for soil rehabilitation after mine activities, and

e. Goals and hypotheses of the present work.

**Response**: The manuscript introduction has been recast to reflect the progression you have suggested. Thank you!

**Editor**: 2. Was Fluorescence measured on the Petri dishes or in the field?

**Response**: In Section 2.2.2 Line 7 we have added "in the field.."

**Editor:** 3. In section 2.3.1 you described 12 replicates per site, but you have only 8 samples, please clarify

**Response**: Added to Section 2.3.1 line 3 "…twelve replicates (subsampled from the eight samples)…

**Editor**: 4. Section 3.1 is called Eco-physiological properties of biocrust cyanobacteria. I wonder if Ph, E.C, etc are eco-physiological properties of biocrusts

**Response**: We agree that this term is not quite correct and changed the headings and in text use to 'Biophysical' In Methods 2.2 and results 3.1 the subheading now reads "Biophysical characteristics of biocrusts and cyanobacteria' whose measures include pH, EC, biomass (chlorophyll $a$), C, N and bioavailable N. This also links nicely to the statement in the introduction "Mining disturbance alters the biophysical state of the biocrust community through excavation, crushing, mixing and burial."

**Editor**: 5. Significant differences between soil types/ land units/etc should be included in ALL different figures and tables, this can be easily done by using superscripts letters

**Response**: When considering this request we believe that the p values are the most relevant (stated in results) and it will look messy as we now have meshed two graphs (new Figure 3) where significance is visually obvious (with error bars) in graphs. However, we have added p values to figure descriptions for additional clarity. Also with new Figure 8 annotations would become messy so key p values have been added to descriptions.

**Editor**: 6. I agree with reviewer comment, too many figures are included within the main document, are is it possible to remove some of them from the main text?
**Response**: We have revised the figures as suggested (now down to eight, see below)

**Editor**: 7. Figure comments:
Fig 1 and 2 can be easily combined in one, same for figure 4 and 5
**Response**: We have combined Figures 1 and 3 as they fit together better visually – if this is OK we can improve graphics for final copy.

10  **Editor**: Fig 6, 7 can be a and b part of the same figure
**Response**: We have combined them.

**Editor**: In fig 8, scale values from axes is missing
**Response**: MDS shows relative similarity and does not have a scale

**Editor**: Figure 9 has not been referenced in the text, please move it to the supplement
**Response**: Moved

**Editor**: Figure 10: Is it possible to use different colors for the different species or genera instead of different
20  levels of green?
**Response**: Changed to blue-greens, looks better.

**Editor**: Fig 11: Does it represent different soils units of SMU. Please be consistent with definition and nomenclature
25  **Response**: changed to T2 (stockpile), others are SMU which is correct

**Editor**: Figures and tables format should be homogenized
**Response**: Done

30  **Editor**: Moreover, figure citation order should be revised
**Response**: Done

**Editor**: Finally I would suggest reformatting some of the main conclusions, as some lines are general statements and not conclusions based on your results
35  **Response**: Conclusions revised

[revised manuscript text omitted]

---

## Editor Decision (ED3)

**Microbial Biobanking**
**Cyanobacteria-rich topsoil facilitates mine rehabilitation**

Wendy Williams[1], Angela Chilton[2], Mel Schneemilch[1], Stephen Williams[1], Brett Neilan[3], Colin Driscoll[4]

1. School of Agriculture and Food Sciences, The University of Queensland, Gatton Campus 4343 Australia
2. Australian Centre for Astrobiology and School of Biotechnology and Biomolecular Sciences, University of New South Wales, Sydney, NSW, 2052, Australia
3. School of Environmental and Life Sciences, University of Newcastle, Callaghan, NSW, 2308, Australia
4. Hunter Eco, PO Box 1047, Toronto, NSW, 2283

*Correspondence to*: wendy.williams@uq.edu.au

Dear Editor,

Thank you for your suggestions. Changes as requested with some additional comments have been made below.

Kind regards,

Wendy Williams

Section 1.0

Page 2, Line 10: Paragraphs 1&2 connected with phrase added.

Page 2, Line 18: Phrase added to start of sentence.

Section 2.0

Page 5, Lines 1&2: biocrust description deleted and sentence merged with following paragraph.

Page 5, Line 21: parameters deleted and replaced with SMU (removed brackets).

Page 5, Lines 24-25: Lake Ifould removed from descriptions

Page 7, Line 3: Sentence clarified "In the field this process was completed at least six times adjacent to each sampled location".

Page 7, Line 5: Sentences revised to read: "Significant differences in C and N and C:N between SMUs as well as differences in Chlorophyll *a* and YII between SMUs and the two-year old stockpile were tested by one-way ANOVA's and Tukey post hoc tests (Minitab 18)."

Page 8, Lines 8-10: Samples taken clarified, first sentence of paragraph revised to read: "For genomic profiling of naturally occurring successional biocrust communities, a location adjacent to Site 9 was visually determined to contain Bare, Early (Crust Types 1–2, SMU 1) or Late (Crust Types 4–5, SMU 2, SMU 3) stages of development (Table S2)." And in sentence queried now reads: "For each successional stage (representative of SMUs 1–3)…."

Page 9, Lines 7-8: Analysis between depths clarified, sentence now reads: "Differences in relative abundance between age and depth for stockpiles versus adjacent undisturbed sites were determined by ANOVA's and Tukey's post hoc tests."

Section 3

Page 9, Line 15: Sentences removed as suggested and additional phrases added.

Page 9, Line 20: words removed.

Page 9, Line 22: T2 was only compared to its origin (SMU 3) however I changed the letter to c to indicate its difference as this is also clarified in results anyway.

Page 9, Line 26: words removed.

Page 9, Line 26-29: letters added to Fig. 3 as requested.

Page 10, Lines 1-3: phrases removed and sentence relocated as suggested.

Page 10, Line 11: changed to diversity and richness.

Page 10, Line 15: Sentence moved up.

Page 10, Line 25: Paragraphs merged.

Page 11, Line 10: Sentenced rephrased, words removed end of paragraph.

Pages 11-12, Lines 25 on: First sentence second paragraph deleted and merged with previous one.

Fig. 8 to be revised. We agree this is confusing. No changes to the superscripts have been made yet because we are revising the presentation of these two graphs to simplify/clarify the understanding of the results (email with additional detail to follow this submission). In the revised graphs we will reverse the depths from left to right and provide the superscript letters for significant differences between ages as per your email.

Section 4

Page 13, Line 2: We have reordered the sentences in this paragraph to clearly show the diversity/richness links to microprocesses.

Page 13, Line 19 on: We have moved the paragraph up and linked to stockpile results.

Page 14, Line 16: We have moved this whole paragraph to the beginning of this section.

[revised manuscript text omitted]

---

## Author Response (AR4)

**Microbial Biobanking**
**Cyanobacteria-rich topsoil facilitates mine rehabilitation**

Wendy Williams[1], Angela Chilton[2], Mel Schneemilch[1], Stephen Williams[1], Brett Neilan[3], Colin Driscoll[4]

1. School of Agriculture and Food Sciences, The University of Queensland, Gatton Campus 4343 Australia
2. Australian Centre for Astrobiology and School of Biotechnology and Biomolecular Sciences, University of New South Wales, Sydney, NSW, 2052, Australia
3. School of Environmental and Life Sciences, University of Newcastle, Callaghan, NSW, 2308, Australia
4. Hunter Eco, PO Box 1047, Toronto, NSW, 2283

*Correspondence to*: wendy.williams@uq.edu.au

Dear Associate Editor,

Thank you for your time and effort to provide valuable comments to improve this manuscript. We trust we have addressed them all satisfactorily.

Kind regards,

Wendy Williams and co-authors

**Editor**: requested a revision of the introduction to reflect the following themes (refer to main comments)

    a. Problem
    b. Solutions
    c. Properties and Ecosystem services
    d. Post-mine rehab recommendations
    e. Goals and hypothesis

**Response**: The introduction has been recast to reflect the order recommended

The following issues raised throughout the ms have been addressed and detailed below:

**Editor:** Soil and biocrust type are not considered together in analysis…why?

The editor has enquired if there was a reason why we haven't included biocrust type with soil type in our analysis. The first point to make is that the three studies presented here (cyanobacterial community structure and function, sequencing and stockpiles) were all independent studies that were linked in the common goal but done separately. In addition, the scope of this research was focused on cyanobacteria that we have collectively called biocrusts however preliminary investigations defined these biocrusts into different successional stages (similar to Budel et al. 2009). Although these were identified across the three soil management units (SMU) aside from SMU1 they were reasonably homogenously distributed across the landscape. A characteristic of these biocrusts (SMU2 and SMU3) was that they contained a large number of lichens and

some mosses. However, due to high-level disturbance associated with the mining process it was acknowledged that cyanobacteria would be the early colonisers and lichens would be unlikely to survive. For this reason, J-A mine requested a focused study of cyanobacteria that underlined their role in terms of succession (i.e. early colonisers, soil stabilisers), nutrient cycling and their survival in topsoil stockpiles.

5 Furthermore, in relation to the sequenced biocrusts for bare, early and late successional stages, these were samples located at one site only not for each sample location. Therefore, we suggest that we could not use this data to apply to the ten sites data.

**Methods**

**P4, L16-19** All these points should be clearly discussed (I miss some discussion about goals a) and all these questions need to be solved in the conclusions

10 Response: We have recast the aims and hypotheses to clearly reflect the goals of this research and further addressed them directly in the discussion.

**P5, L14** These are not already defined, please define them

**Response**: We have recast this paragraph to clarify the sampling strategy, vegetation and locations within the SMU with details shown in Table S1 and S2 and Figures 1 and 2.

15 "At J-A the landscape has been characterised into three distinct soil types that were associated with vegetation communities identified as soil management units (Doudle et al., 2011; Hou and Warland, 2005). Site vegetation associations are described as follows: SMU 1 – Red Mallee: *Eucalyptus oleosa* ssp. *oleosa* = open Mallee/Myall woodland; SMU 2 – Chenopod Shrubland: *Maireana sedifolia* and *Atriplex vesicaria*; SMU 3 – Western Myall: *Acacia papyrocarpa Maireana sedifolia* = open Myall woodland Site 1 occurs in a transition between SMU 2 and

20 SMU 3 but was treated as most like SMU 2 (also see Table S1). Soil management units were summarised as: SMU 1 – deep calcareous yellow sands associated with dune ridges; SMU 2 – shallow calcareous sandy loams; and SMU 3 – deep calcareous sandy loam (Table S1). In the first place, sample site locations (Fig. 1) were selected based on these soil management units (SMU) and a two-year old stockpile (Table S1). Secondarily, sites within these parameters (SMU) were selected for the subsequent detailed studies of cyanobacterial succession

25 and its resilience to longer-term stockpiling (Fig.2 and Section 2.4).

The biocrusts at J-A had been previously classified into three successional stages representative of the five biocrust types found growing across the landscape (Table S1, S2) (Doudle et al., 2011). Types 1–2 are light coloured, patchy, thin, and fragile cyanobacterial crusts corresponding to early stages of development; Type 3 are well established cyanobacterial crusts with establishment of some mosses and lichens corresponding to intermediate stages of development; Types 4–5 biocrusts

30 are well established with cyanolichens and/or green algal lichens and mosses corresponding with late stages of development (additional descriptions available in supplementary Table S1). As well as the three main SMU sites we also sampled a dry salt lake (Lake Ifould) to provide information on cyanobacterial species adapted to saline conditions, similar in nature to ground water used in mine operations. In this study the term biocrust covers whole crust samples that incorporated lichens

cyanobacteria and mosses in varying proportions however, the cyanobacterial component of this crust was the focus in terms of our polyphasic approach to community structure, succession and its biophysiochemical properties."

**P5, Section 2.2** As you have two factors (SMU and crust type, a two-way anova is necessary?

**Response**: Refer to detailed explanation above. We did not use crust type as a factor as it was not specifically identified when sampling (for this project). We used prior classifications to provide an informative backdrop for decision making in relation to the sampling strategy employed in this study.

**P5, L30** please define these sites

**Response**: Refer to above revision and additional descriptions now included

**P6, Section 2.2.2** statistical analysis for this section are still missing

**Response**: statistical methods added

**P6, L6** half petri dish or less? Please clarify

**Response**: Clarified in text

**P6, L11-14** was it done before other analysis (before disturbance), or was done based on a subsample, please clarify

**Response**: Clarified in text

**P6, L15-19** was it done before other analysis (before disturbance)? or in different petri dishes

**Response**: Clarified in text

**P6, L21** I will suggest to include that it was done on the field at this point. When it was measured? during the field sampling campaign or in a different date.

**P6, L31** before or after other measurements?

**Response**: Section 2.2.2 has been revised to clarify the above-mentioned points

**P8, L28** (last sentence) differences in what?

Response: this sentence updated to what differences

**Results**

**P9, Section 3.1** Here you only analysed differences between SMU, what about different biocrust types, as some different biocrust types in same SMU may have larger diff than similar crust types at different SMU, it is necessary to include both factors in your analysis

**Response**: Please see earlier explanation regarding biocrust types vs SMU

**P9, L6** significant or no?

These details are now included in text and in the table

**P9, L10 (also Figure 3)** see my comment on figure 3, in a similar way as I suggested on table 3, significant differences should be added in all figures and tables where you performed an analysis of differences between classes. This can be done by using superscripts

e.g. a for the group with higher values, b, c, etc

When to classes do not show significant differences, then they have same superscript

letter should be added in the same way I suggested on table 2

**Response**: done

**Table 2** this can be solved by adding letters a, b and c for the groups with significant differences as super index

5   SMU3 a SMU2 a, as no significant differences between them are found, SMU1 b (significantly lower values than SMU 2

and 3)

**Response**: the letters have been added where required

**Table 4** define 'Permdisp'

**Response**: The term Permdisp is defined in the methods where it is first mentioned.

10   We have included Permdisp definition in legend in Table 4

[revised manuscript text omitted]

---

## Author Response (AR5)

**Microbial Biobanking**
**Cyanobacteria-rich topsoil facilitates mine rehabilitation**

Wendy Williams[1], Angela Chilton[2], Mel Schneemilch[1], Stephen Williams[1], Brett Neilan[3], Colin Driscoll[4]

1. School of Agriculture and Food Sciences, The University of Queensland, Gatton Campus 4343 Australia
2. Australian Centre for Astrobiology and School of Biotechnology and Biomolecular Sciences, University of New South Wales, Sydney, NSW, 2052, Australia
3. School of Environmental and Life Sciences, University of Newcastle, Callaghan, NSW, 2308, Australia
4. Hunter Eco, PO Box 1047, Toronto, NSW, 2283

*Correspondence to*: wendy.williams@uq.edu.au

Dear Associate Editor,

We appreciate your additional suggestions and editing for this manuscript and have addressed each area detailed below. We thank you for your time and input to improve this submission.

Kind regards,

Wendy Williams et al.

Editor:

INTRODUCTION

I suggested to work a little bit more on the introduction as it is still not fully connected.

Response:

We have revised the introduction and focused on your comments to connect the mining rehabilitation process and problems with the role of biocrusts.

Editor:

M&M

Was microscopy done based on same samples used for physicochemical analysis?

not clear in the text.

Response:

We have clarified this in the text

Editor:

RESULTS

Check how significant differences are presented in tables and figures

Figure 8 still not clear, what are you comparing. Moreover, all treatments should include letters

Caption should be something like:

What mean age on undisturbed samples means?

Response:

We have clarified Figure 8 in the positioning of the graphs (stockpile first) and revised the caption to reflect this. (Mean) age is just the comparison to the stockpile age so the reader knows where the sample came from (i.e. adjacent to which stockpile). However we have removed the word mean as it was not relevant.

Editor:

Based on figure, section 3.4 still confusing for an external reader, I would like to see a clearer version with a well define story line

Response:

We have made further changes to this section to make it more succinct.

Editor:

DISCUSSION

I also suggest changing first subsection to include the link between cyanobacterial community structure, nutrient cycling and soil stabilization, as this last part is missing within the discussion

Response:

We have further revised the discussion (in full) and linked the community structure to stabilisation and nutrient cycling (refer to paragraph 1 in the discussion below):

"This research has demonstrated cyanobacteria to be a key component of soil microbial communities at J-A. These were compositionally diverse topsoil microbiomes that substantially contributed to the Myall-chenopod landscape. We had hypothesised that cyanobacteria would be central to soil micro-processes, and this was strongly supported by extensive species richness and diversity values. At J-A cyanobacteria contributed to soil structure and function during the early developmental stages of the biocrust. Photosynthesis drove the productivity and growth of the biocrust that initiated carbon and nitrogen cycling and resulted in increases in soil nutrient concentrations right where vascular plants might use them. The results have demonstrated how these micro-processes provide a strong foundation for the restoration of soil function. Similarly, in southwestern Queensland and northern Australia cyanobacterial species richness was strongly linked to increased bioavailable nitrogen and carbon uptake (Büdel et al., 2018; Williams et al., 2018; Williams and Eldridge, 2011). Elsewhere, multiple studies have demonstrated the high value of biocrust attributes as drivers of soil micro-processes that restore soil function (e.g. (Barger et al., 2016; Belnap and Eldridge, 2001; Bowker et al., 2014; Büdel et al., 2009; Chilton et

al., 2017; Chiquoine et al., 2016; Weber et al., 2016)."

Editor:

Conclusions:

I also recommended to rewrote in order to ask main objectives presented:

A general conclusion based on your main objective "to determine whether shallow biobanks of cyanobacterial-enriched topsoil would facilitate the recovery of essential soil microprocesses when re-spread following mine disturbance"

and 2 specific conclusion based on specific objectives:

1) Define the cyanobacterial community structure with a special focus on species that drive early colonization, nutrient cycling and soil stabilization

2) to examine the effects of stockpiling topsoil on cyanobacterial resilience to crushing and burial and their recovery following spreading of topsoil back across mined land.

Response:

Conclusion has been fully revised to reflect comments.

[revised manuscript text omitted]

---

## Author Response (AR7)

Dear Editor,

We have amended Figure 8 and made all the other changes as suggested.

Thank you for your assistance to improve this manuscript.

Kind regards,

Wendy Williams and co-authors

[revised manuscript text omitted]